# HP1 proteins compact DNA into mechanically and positionally stable phase separated domains

Madeline M Keenen[1,2], David Brown[3], Lucy D Brennan[1], Roman Renger[4,5], Harrison Khoo[6], Christopher R Carlson[2,7], Bo Huang[1,3,8], Stephan W Grill[4,9], Geeta J Narlikar[1]*, Sy Redding[1,10]*

[1]Department of Biochemistry and Biophysics, University of California, San Francisco, San Francisco, United States; [2]Tetrad Graduate Program, University of California, San Francisco, San Francisco, United States; [3]Department of Pharmaceutical Chemistry, University of California, San Francisco, San Francisco, United States; [4]Max Planck Institute of Molecular Cell Biology and Genetics, Dresden, Germany; [5]German Center for Neurodegenerative Diseases (DZNE), Bonn, Germany; [6]Department of Mechanical Engineering, Johns Hopkins University, Baltimore, United States; [7]Department of Physiology, University of California, San Francisco, San Francisco, United States; [8]Chan Zuckerberg Biohub, San Francisco, United States; [9]Cluster of Excellence Physics of Life, Technische Universität Dresden, Dresden, Germany; [10]Marine Biological Laboratory, Woods Hole, United States

*For correspondence:
Geeta.Narlikar@ucsf.edu (GJN);
syeugene.redding@ucsf.edu (SR)

Competing interests: The authors declare that no competing interests exist.

**Abstract** In mammals, HP1-mediated heterochromatin forms positionally and mechanically stable genomic domains even though the component HP1 paralogs, HP1$\alpha$, HP1$\beta$, and HP1$\gamma$, display rapid on-off dynamics. Here, we investigate whether phase-separation by HP1 proteins can explain these biological observations. Using bulk and single-molecule methods, we show that, within phase-separated HP1$\alpha$-DNA condensates, HP1$\alpha$ acts as a dynamic liquid, while compacted DNA molecules are constrained in local territories. These condensates are resistant to large forces yet can be readily dissolved by HP1$\beta$. Finally, we find that differences in each HP1 paralog's DNA compaction and phase-separation properties arise from their respective disordered regions. Our findings suggest a generalizable model for genome organization in which a pool of weakly bound proteins collectively capitalize on the polymer properties of DNA to produce self-organizing domains that are simultaneously resistant to large forces at the mesoscale and susceptible to competition at the molecular scale.

## Introduction

Compartmentalization of the eukaryotic genome into active and repressed states is critical for the development and maintenance of cell identity (*Becker et al., 2016*; *Maison and Almouzni, 2004*). Two broad classes of genome compartments are heterochromatin, which contains densely packed DNA regions that are transcriptionally repressed, and euchromatin, which contains physically expanded DNA regions that are transcriptionally active (*Allshire and Madhani, 2018*; *Heitz, 1928*; *Saksouk et al., 2015*). A highly conserved type of heterochromatin involves the interaction of proteins from the heterochromatin Protein 1 (HP1) family with chromatin that is methylated on histone H3 at lysine 9 (*Bannister et al., 2001*; *Eissenberg et al., 1990*; *James and Elgin, 1986*; *Lachner et al., 2001*). In addition to repressing transcription, this type of heterochromatin also plays

critical roles in chromosome segregation and in conferring mechanical rigidity to the nucleus (*Allshire and Madhani, 2018*; *Stephens et al., 2019*).

From investigations of chromatin in cells, it is not immediately obvious how to connect the biophysical properties of HP1 proteins to the diverse roles of HP1-mediated heterochromatin. Heterochromatin domains are typically found to be statically positioned within the nucleus for several hours, held separate from euchromatin (*Gerlich et al., 2003*; *Marshall et al., 1997*). Yet, these domains can also be rapidly disassembled in response to environmental and developmental cues (*Cheutin and Cavalli, 2012*; *Dion and Gasser, 2013*; *Kind et al., 2013*). The finding that HP1 molecules in these domains exchange within seconds provides some insight into how these domains can be dissolved, because competing molecules would be able to rapidly displace HP1 proteins from DNA (*Cheutin et al., 2003*; *Festenstein et al., 2003*). However, such models raise the fundamental question of how HP1 molecules, which are dynamic on the order of seconds, enable chromatin states that are stable on the order of hours, and further how these states can resist the forces exerted on chromatin in the cell.

The mammalian genome contains three HP1 paralogs: HP1α, HP1β, and HP1γ. While the three paralogs show a high degree of homology, they are associated with distinct biological roles (*Canzio et al., 2014*; *Eissenberg and Elgin, 2014*). For example, HP1α is mostly associated with gene repression and chromosome segregation (*Allshire and Madhani, 2018*; *Canzio et al., 2014*; *Eissenberg and Elgin, 2014*), HP1β plays both gene activating and gene repressive roles (*Allshire and Madhani, 2018*; *Canzio et al., 2014*; *Eissenberg and Elgin, 2014*), and HP1γ is more often associated with promoting transcription (*Allshire and Madhani, 2018*; *Canzio et al., 2014*; *Eissenberg and Elgin, 2014*). These observations raise the question of how small differences at the amino acid level give rise to distinct biophysical properties that direct the different functions of the HP1 paralogs.

Some of the questions raised above have been investigated in vitro. For example, it has been shown that HP1 proteins are sufficient to bind to DNA and chromatin and to provoke their robust condensation (*Azzaz et al., 2014*; *Canzio et al., 2011*; *Kilic et al., 2018*; *Kilic et al., 2015*; *Larson et al., 2017*; *Meehan et al., 2003*; *Mishima et al., 2013*). These experiments have led to a model where HP1 molecules, by means of multiple contacts, condense and staple chromatin structures in place. Furthermore, and consistent with cellular measurements, HP1 molecules also exhibit weak affinity for chromatin in vitro (*Canzio et al., 2013*; *Canzio et al., 2011*; *Kilic et al., 2015*). Recent findings of phase-separation behavior by HP1 proteins provide an added perspective to the questions above (*Larson et al., 2017*; *Sanulli et al., 2019*; *Strom et al., 2017*; *Wang et al., 2019*). Specifically, the human HP1 protein, HP1α was shown to undergo liquid-liquid phase separation (LLPS) in vitro when phosphorylated and in combination with DNA (*Larson et al., 2017*). Parallel studies showed that the *Drosophila* HP1 protein, HP1a, also forms phase-separated condensates in vivo (*Strom et al., 2017*). In contrast, HP1β cannot undergo LLPS in vitro upon phosphorylation or in combination with DNA, but can be recruited to liquid phases of modified chromatin (*Larson et al., 2017*; *Wang et al., 2019*). The biophysical interactions that give rise to in vitro LLPS are consistent with the in vivo observations of low-affinity binding and chromatin condensation by HP1α. The weak interactions underlying HP1-mediated LLPS also provide an attractive rationale for the rapid invasion and disassembly of heterochromatin. However, such an LLPS-based model does not easily explain the mechanical and temporal stability of chromatin domains.

A recent study has implied that HP1-mediated heterochromatin in cells does not exhibit liquid-like phase-separated behavior (*Erdel et al., 2020*). This conclusion was based on definitions derived from the material properties of a subset of LLPS systems in vitro, such as impermeable boundaries and concentration buffering. However, these properties do not translate simply from in vitro to in vivo settings as condensates in cells span a diversity of protein environments and solvation conditions that will vary the nature of their boundaries and partitioning of nuclear material. Such narrow definitions are not generally applicable and fail to capture the nature of several types of condensates (*McSwiggen et al., 2019a*; *Riback et al., 2020*). Specifically, for condensates that involve DNA, there are additional constraints that arise from the properties of long polymers that do not scale in a straightforward way from smaller systems. These important considerations underscore the need to move beyond simple definitions and better understand the different and sophisticated ways in which condensates play biological roles.

Here, using a combination of ensemble and single-molecule methods, we uncover the molecular basis of intramolecular DNA compaction by HP1α and the molecular determinants that give rise to HP1α-induced phase separation. In doing so, we investigate the role of DNA in condensates, both as a binding partner for HP1α and as a long polymer with unique organizational constraints. We show that condensates of HP1α and DNA are maintained on the order of hours by HP1α binding that is dynamic on the order of seconds. We find that the central disordered region of HP1α is sufficient to enable LLPS with DNA, and that the additional disordered regions regulate the activity of this central region. These results are then leveraged to uncover intrinsic biophysical differences across the three human HP1 paralogs. Finally, we show that the HP1α-DNA condensates are resistant to mechanical disruption by large forces and yet can be readily dissolved by HP1β. Overall, our results uncover specific biophysical properties of each HP1 paralog in the context of DNA that have general implications for interpreting and understanding the behaviors and functions of HP1 in the context of chromatin.

## Results

From previous work, we have found that HP1α shows the most robust phase-separation and DNA compaction abilities of all of the HP1 paralogs (*Larson et al., 2017*). We therefore first used HP1α and DNA as a model system to dissect the steps involved in DNA compaction and phase-separation and to study the material properties of the resultant phases. We then carried out structure-function analysis on HP1α to understand how different regions of HP1α contribute to phase-separation. The results from these studies provided a well-defined biophysical framework within which to (i) compare the activities of HP1β and HP1γ, and (ii) understand how HP1β and HP1γ impact the phase-separation activities of HP1α. Finally, throughout we compare our observations of HP1-DNA condensates with prevailing views of the expected behavior of condensates.

### HP1α binds DNA globally but compacts DNA locally

We have previously shown that HP1α rapidly compacts long stretches of DNA (*Larson et al., 2017*). To understand the mechanism of DNA compaction, we have leveraged a single molecule DNA curtain approach (*Figure 1A*; *Greene et al., 2010*). In this assay, ~50 kbp molecules of DNA from bacteriophage λ are fixed to the surface of a microfluidic flowcell via a supported lipid bilayer. Visualization of DNA is achieved by labeling with the intercalating dye YOYO-1 (*Figure 1B–D,F*). HP1α is then pulsed into the flowcell, driving rapid DNA compaction (*Figure 1B,D–F*, *Figure 1—figure supplement 1A–C*). Previously, we showed that HP1α-induced DNA compaction is an electrostatically driven process that proceeds by first concentrating DNA at the free end, and then rapidly and sequentially incorporating upstream DNA into a single condensate (*Figure 1B*; *Larson et al., 2017*). We validated that compaction occurs at the free end by labeling the untethered end of the DNA with a fluorescent dCas9 (*Figure 1C–E*).

To further understand how HP1α compacts DNA, we directly visualized fluorescently labeled HP1α binding to DNA during compaction. Surprisingly, we found that HP1α binds uniformly along DNA, incorporating into both the compacted and uncompacted regions (*Figure 1E–L*). We observed a linear increase in fluorescence due to HP1α binding on uncompacted DNA (*Figure 1I*). And by comparison, we found that HP1α incorporates into compacted DNA at the same rate as on uncompacted DNA at 50 µM HP1α, and moderately faster into the compacted DNA at 5 µM HP1α (*Figure 1J–L*). We conclude that compacted DNA states are not inaccessibly compacted, but rather continue to support ingress and egress of HP1α from solution.

We considered two possibilities to explain how global binding would manifest in local compaction. In the first possibility, HP1α binding is coupled to bending of the binding site. In such a case, the cumulative effect of multiple HP1α binding events would appear as a scrunching of the DNA fiber that would be evident in the fluorescence HP1α or DNA signal. However, we observe no appreciable increase in the YOYO-1 signal on non-compacted DNA during compaction (*Figure 1—figure supplement 2A*). In addition, when we directly label HP1α instead of the DNA, we observe a linear increase in fluorescence on the uncompacted segment of the DNA (*Figure 1I*) consistent with HP1α binding in the absence of appreciable DNA bending of the binding site. Whereas a supralinear increase in HP1α fluorescence would be expected if the fluorescent signal was the product of HP1α association and increased local DNA density as a result of bending.

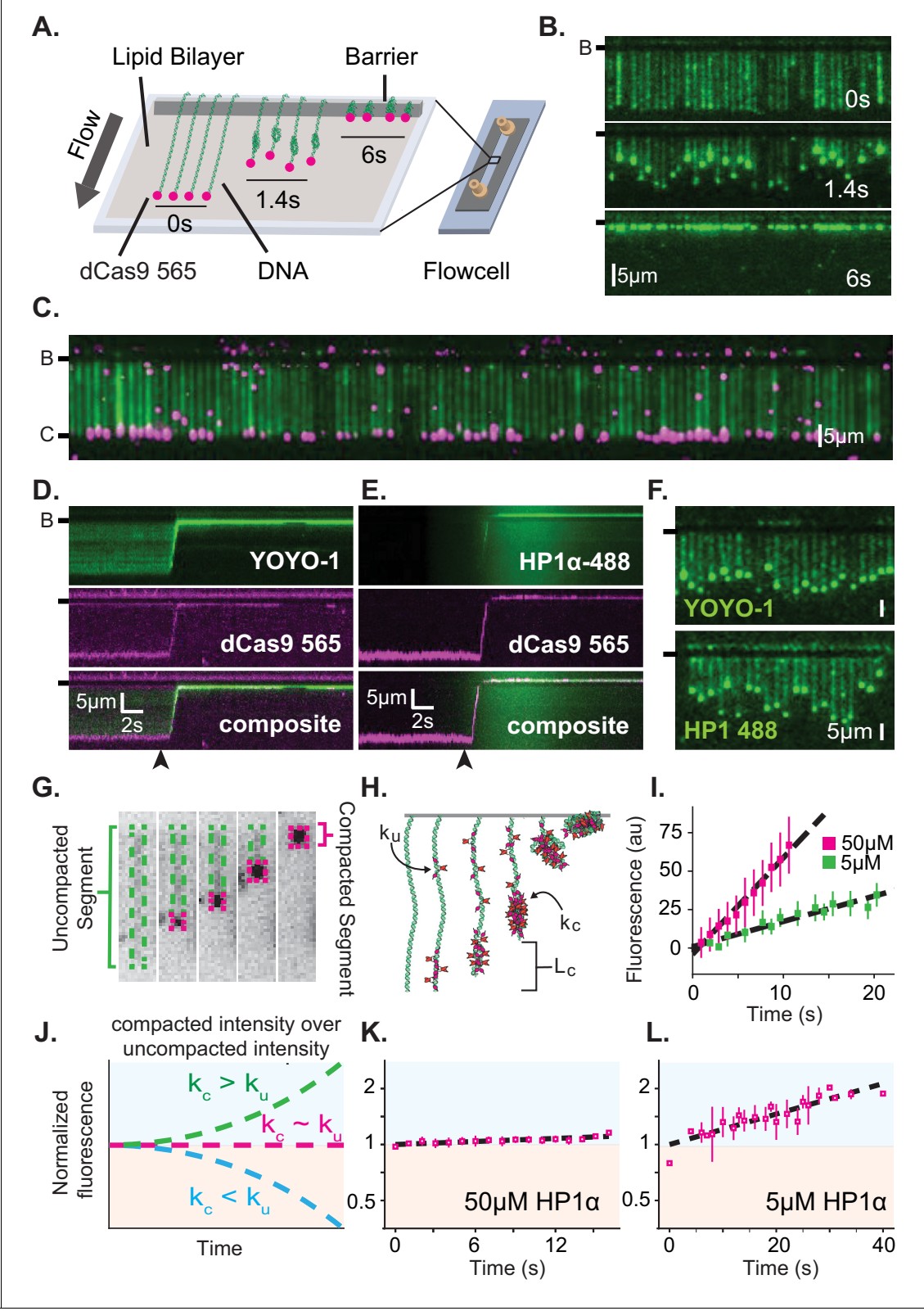

**Figure 1.** Characterization of DNA compaction by HP1α. (**A**) Cartoon of the DNA curtains assay showing compaction of DNA. (**B**) Timestamped images of DNA labeled with YOYO-1 undergoing compaction by 50 μM HP1α (unlabeled) shown before, during, and after compaction. (B-) or (-) specifies location of the barrier. (**C**) DNA curtain end-labeled with fluorescent dCas9 (C-). The dCas9 is targeted to a site 750 bp from the untethered end of the DNA. (**D** and **E**) Kymograms of DNA compaction by 50 μM HP1α. (**D**) DNA labeled with YOYO-1 (top), dCas9-565 (middle), and composite image

*Figure 1 continued on next page*

*Figure 1 continued*

(bottom). (E) HP1α−488 (top), DNA labeled with dCas9-565 (middle), and composite image (bottom). Arrowheads represent estimated time of protein injection. (F) Still images during DNA compaction of either DNA labeled with YOYO-1 (top) or HP1α−488 (bottom). (G) A DNA molecule undergoing compaction by HP1α specifying the uncompacted segment (green) and compacted segment (magenta). (H) Cartoon of HP1α compacting DNA over time. $L_c$ is the length of compacted DNA, $k_u$ is the rate of fluorescence increase for the uncompacted DNA segment, and $k_c$ is the rate fluorescence increase for the compacted DNA segment. See Materials and methods for more information. (I) Fluorescence increase of HP1α−488 on uncompacted DNA. N = 25 for both concentrations, error bars represent standard deviations. (J) Cartoon showing potential results from normalizing the fluorescence of the compacted segment by that of the uncompacted segment. (K and L) Measured normalized compacted HP1α fluorescence relative to uncompacted HP1α. N = 25 for both concentrations, error bars represent standard deviations.

The online version of this article includes the following figure supplement(s) for figure 1:

**Figure supplement 1.** DNA compaction at different HP1α concentrations.

**Figure supplement 2.** Fluorescence conservation and tracking DNA compaction by HP1α.

In the second possibility, HP1α molecules could trap naturally occurring DNA fluctuations by binding to multiple distal DNA sites simultaneously, or through the interactions of two or more HP1α molecules pre-bound to distal DNA sites. Indeed, the rapid and constant speed of DNA compaction against buffer flow (47 kbp/s at <1 pN for 50 µM HP1α) suggests that HP1α capitalizes upon DNA fluctuations that bring linearly distal segments of DNA together (*Figure 1—figure supplement 1C*; *Baumann et al., 2000*; *Ostrovsky and Bar-Yam, 1994*). Such a model then explains why the initiation of compaction is localized to the untethered end of the DNA: the lower tension at the untethered end allows for a larger number of DNA conformations that bring distal regions of the DNA into close proximity. HP1α is then able to trap these conformations leading to increased inclusion into the growing condensate either through HP1α-DNA or potentially through HP1α-HP1α interactions. The uniform binding of DNA by HP1α may additionally result in DNA that is easier to compact by altering the effective persistence length of the coated polymer.

From the results above, we identify three regulatable steps of HP1α-DNA condensation: local assembly of HP1α along DNA prior to DNA condensation, initiation of DNA compaction through capturing of lateral DNA fluctuations, and progression of DNA compaction through inclusion of uncompacted DNA into the growing condensate via HP1α-DNA and HP1α-HP1α interactions. As described in the discussion, nucleosomes and other nuclear factors will modulate each of these steps to further regulate DNA compaction.

## Condensate formation is more sensitive to the concentration of HP1 than of DNA

HP1α behaviors that result in DNA compaction at the single molecule level will also produce meaningful effects at the meso-scale. To further uncover the molecular details of how HP1α organizes DNA, we generated a phase diagram of HP1α-DNA condensation using short (147 bp) double-stranded DNA oligomers (*Figure 2A*). The length of the DNA (near the persistence length for B-DNA) was constrained to study the role of HP1α-DNA and potential HP1α-HP1α interactions while minimizing extensive polymer behaviors of DNA. At the conditions these experiments were performed (70 mM KCl, 20 mM HEPES pH 7.5, 1 mM DTT), HP1α remains soluble even at exceedingly high concentrations (400 µM) (*Figure 2A*, bottom right panel). However, in the presence of DNA, HP1α readily condenses into concentrated liquid phase-separated material (*Figure 2A*) indicating the formation of a network of weak interactions interconnecting HP1α and DNA molecules. Such interactions are consistent with HP1α's ability to capture and stabilize distal segments of DNA leading to DNA compaction as discussed in the previous section.

One way to quantify the phase-separation capability of a molecule is through measurement of its critical concentration. Empirically, the critical concentration is defined as the concentration of the molecule above which the system separates into two phases. Theoretically, this transition occurs at the concentration at which the collective weak interactions of the system pay the entropic cost of de-mixing. This means that anything that affects the strength or number of interactions will also shift the critical concentration. For example, raising the concentration of monovalent salt will weaken electrostatically driven HP1-DNA interactions and increase the critical concentration. In a two-component system, such as HP1α and DNA, each component may contribute differentially to

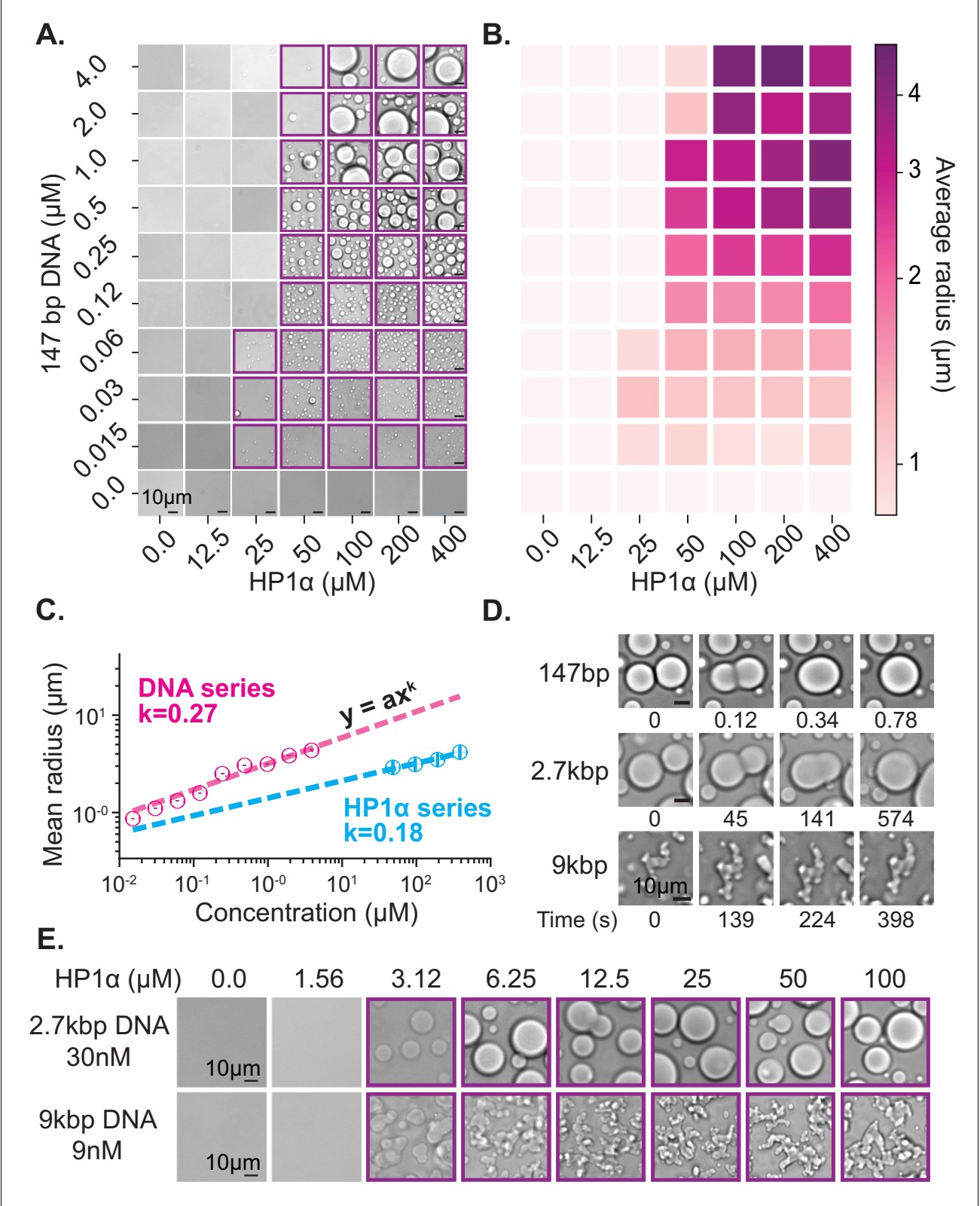

**Figure 2.** Characterization of HP1α-DNA condensate formation. (A) Bright-field images of mixtures of HP1α and 147 bp DNA. (B) Heat map of the average radius of condensates for each condition in (A). (C) Average condensate radius for 1 μM 147 bp DNA plotted against HP1α concentration (cyan) or 100 μM HP1α plotted against 147 bp DNA concentration (magenta) and fit to a power law, error bars (obscured) represent the SEM. (D) Time

*Figure 2 continued on next page*

*Figure 2 continued*

stamped brightfield images of 100 μM HP1α and 147 bp, 2.7 kbp, or 9 kbp DNA depicting fusion and coalescence behavior. (E) Brightfield images of HP1α with either 30 nM 2.7 kb DNA (top) or 9 nM 9 kbp DNA (bottom). Throughout, purple boxes indicate presence of condensates.

The online version of this article includes the following figure supplement(s) for figure 2:

**Figure supplement 1.** Exceedence probability of HP1α-DNA condensates.
**Figure supplement 2.** Characterization of HP1α condensates.

condensation, and measuring the critical concentration of each component can provide insights into how the two components interact to form condensates.

First, we estimated the critical concentration of HP1α necessary to induce phase separation to be ~50 μM in the presence of 147 bp DNA at concentrations ranging from 0.125 to 4 μM (*Figure 2A*). However, above this critical concentration of HP1α, we were unable to measure a corresponding critical concentration for DNA. Rather, lowering the DNA concentration resulted in a continuous reduction in the average size of observed HP1α-DNA condensates instead of a sharp disappearance (*Figure 2A–C*, *Figure 2—figure supplement 1A*). Thus, we conclude the critical concentration of HP1α is largely invariant of DNA concentration. In fact, even at very large ratios of HP1α to DNA (5000:1, *Figure 2—figure supplement 2A*), we still observe condensates.

The apparent dissociation constant for HP1α interactions with ~60–200 bp DNA ranges from 0.3 to 10 μM (*Nishibuchi et al., 2014*; *Figure 2—figure supplement 2D*) which means, for most of the conditions tested here where we observe macroscopic droplets, we expect that nearly all DNA molecules are fully bound by HP1α. Once a collection of HP1α molecules coat a single DNA, that DNA molecule and its associated HP1α can, on average, act as a single highly valent molecule, or proto-condensate, that acts as a liquid building block and aggregates with other HP1α-DNA proto-condensates as they encounter one another in solution (*Kilian et al., 1997*). It is helpful to recall that DNA regions already bound by HP1α were readily incorporated into condensates in our curtain assay, and the same biophysical considerations above also apply here. Specifically, we expect that condensate formation and growth are dependent on the concentration of HP1α and are the result of either higher order HP1α oligomerization or molecular rearrangements along DNA oligomers interacting in trans.

The ensuing aggregation process—proto-condensates clustering into large macroscopic condensates—should result in condensates sizes distributed according to a power law; where the power is set by molecular rates of diffusion and absorption (*Brangwynne et al., 2011*; *Vicsek and Family, 1984*; *Weitz and Lin, 1986*). Specifically, this result comes about because increasing the HP1α or DNA concentration increases the rate of formation and total number of proto-condensates, which increases their encounter frequency in solution accelerating the process of diffusion-driven aggregation. To test this hypothesis, we measured the average radius of condensates as a function of DNA and HP1α concentration (*Figure 2B–C*, *Figure 2—figure supplement 1A*, *Figure 2—figure supplement 2B*). We find the average droplet size versus concentration of both DNA and HP1α is in fact well described by a power law (*Figure 2C*), further connecting the formation of macroscopic liquid droplets to the microscopic processes of aggregation and DNA compaction.

HP1α-HP1α oligomerization may be a driving force in the HP1α-DNA aggregation we observe. Notably, at 40 mM KCl, high concentrations (~400 μM) of HP1α can undergo LLPS in the absence of DNA (*Figure 2—figure supplement 2C*). However, under similar ionic conditions to those used in the bulk of this study (75 mM KCl), and in the absence of DNA, the critical concentration for HP1α to exhibit LLPS is greater than 800 μM, and HP1α predominantly exists as a dimer (*Larson et al., 2017*). Based on these findings, we propose that HP1α has the ability to form higher order oligomers by itself, but that this is a salt-dependent process that is enhanced by the presence of DNA.

While our data are consistent with HP1α-DNA binding promoting higher order HP1α oligomerization, at the same time, prior work suggests that the interface involved in HP1α-HP1α interactions following phosphorylation overlaps with the interface involved in HP1α-DNA interactions. If HP1α oligomerization is a key factor driving condensation, we then predict that as DNA concentration is increased, eventually HP1α-DNA interactions will outcompete HP1α-HP1α interactions, resulting in a loss of condensation. However, an alternative, compatible explanation suggests that as DNA concentration is increased, each DNA molecule is no longer bound by a sufficient amount of HP1α to

create a productive proto-condensate or stabilize macroscopic condensates. Consistent with both of these expectations, at concentrations approaching equimolar ratios of HP1α to DNA binding sites (assuming 60 bp per HP1α dimer-binding site (Materials and methods)—At 50 μM HP1α and 2–4 μM 147 bp DNA) droplet formation is abrogated (*Figure 2A–B*, *Figure 2—figure supplement 2A*).

Overall, the behavior of HP1α and DNA in this condensation assay is consistent with the compaction process we measure in our single molecule assay, and ultimately our results demonstrate that DNA and HP1α play qualitatively different roles in the formation of the HP1α-DNA condensates. In both assays, at suitable HP1α concentrations, HP1α condenses locally around a single DNA molecule. In the curtains assay, DNA is then compacted through lateral HP1α-DNA and possible HP1α-HP1α interactions in cis, whereas in the droplet assay, HP1α and DNA collectively condense into proto- and macroscopic condensates in trans. Additionally, both assays suggest that HP1α engaged with a single DNA molecule samples the same biophysical states as HP1α molecules contained within compacted structures and large macroscopic phases. However, an important difference between these two assays is the length of DNA. We observe robust DNA condensation on curtains at concentrations lower than the critical concentration for HP1α-DNA LLPS measured here on short DNA oligomers (*Figure 1—figure supplement 1B–C*, *Figure 2A*), indicating changes in DNA length will affect the formation of condensates. Moreover, we expect that as DNA length is increased, the conformational constraints and increased binding site availability of longer polymers will also have profound effects on the formation and material properties of HP1α-DNA condensates.

## The length of the DNA affects critical concentration and viscosity

The above studies were designed to minimize the contributions of DNA polymer length to allow us to investigate how multivalent interactions between HP1α and DNA promote the formation of condensates. At the scale of individual HP1α molecules, these multivalent interactions have many similarities to the types of multivalent interactions described in liquid-liquid phase-separating protein-protein and protein-RNA systems (*Jain and Vale, 2017*; *Li et al., 2012*). However, at genomic scales, two features of HP1α-DNA condensates are expected to diverge from other commonly studied phase-separating systems. First, the size disparity between DNA in the nucleus and HP1α is several orders of magnitude. Therefore, neither the valency nor concentration of DNA is expected to be limiting for HP1α condensation in the nucleus. In contrast, conditions are possible in the cell where the valency and concentration of scaffolding RNA molecules or client proteins are in short supply. Second, the length of genomic DNA will have profound bulk-level effects on condensate viscosity and morphology that will be distinct from other phase separating biological mixtures. Consequently, current definitions need to be modified when discussing phases formed in the context of HP1 proteins to explicitly include the polymer behavior of DNA. Toward this goal, we next investigated the effects of increasing DNA length on HP1α-DNA condensates. We expected to observe two results: lower critical concentrations of HP1α necessary to induce condensation due to increases in DNA valency and increases in bulk viscosity resulting in subsequent changes to the shapes of condensates.

Upon increasing the size of linear DNA co-incubated with HP1α from 147 bp to 2.7 kbp, we observed an order of magnitude decrease in the critical HP1α concentration required to induce LLPS (50 μM to 3 μM) (*Figure 2A,E*). This reduction drops the critical concentration to within the estimated range of HP1α concentrations in vivo (1–10 μM) (*Lu et al., 2000*; *Müller et al., 2009*). This result is consistent with the roughly one order of magnitude increase in estimated HP1α-binding sites from ~2 to ~45 per DNA molecule (*Rubinstein and Colby, 2003*). Consistent with the electrostatic nature of HP1α-DNA interactions, increasing the KCl concentration from 70 to 150 mM increases the critical concentration back to ~50 μM for 2.7 kbp DNA (*Figure 2—figure supplement 1*). Conversely, at 70 mM KCl, increasing the DNA length from 2.7 kbp to 9 kbp did not lead to an additional decrease in the critical concentration of HP1α (*Figure 2E*). This apparent lower limit for the critical HP1α concentration at 70 mM KCl is coincident with prior measurements of the HP1α-HP1α dimerization constant under the same conditions, raising the possibility that dimerization plays an essential role in phase-separation (*Larson et al., 2017*). Alternatively, this result may indicate that increasing the length of DNA beyond a certain size does not correspond to further increases in valency because the added DNA segments are distal enough to behave independently. In our single molecule assay, we observed HP1α-induced DNA compaction at concentrations as low as 500 nM (*Figure 1—figure supplement 1B–C*). However, the rate of DNA compaction exhibited by 500 nM

HP1α was roughly 30 times slower than the compaction rate at 5 μM HP1α where we might have predicted only a 10 times slower rate of compaction based on an expected change in the pseudo-first order association rate constant (*Figure 1—figure supplement 1C*). This suggests that HP1α dimerization modestly increases HP1α's on-rate for DNA binding. In addition, the sharp loss of condensates at concentrations where DNA binding and slower DNA compaction still occurs, indicates that dimerization is kinetically upstream of condensate formation and/or affects HP1α-DNA binding parameters, which are not critical during compaction.

In addition to changes in critical concentration, we also observe a marked reduction in the rate of coalescence of HP1α-DNA condensates formed from longer DNA lengths (*Figure 2D*). HP1α-DNA condensates formed with 147 bp DNA rapidly coalesce into spherical structures immediately following fusion (*Figure 2D*). However, increasing the DNA length to 2.7 kbp substantially (>100 times) lengthens the time required for coalescence (*Figure 2D*). Such slower coalescence could be reflective of decreasing surface tension and/or increasing viscosity. It is unlikely that DNA-DNA binding modes contribute to the condensate surface tension. Therefore, we assume that surface tension arises through HP1α-DNA and potentially HP1α-HP1α interactions, which should both be unchanged in character upon increasing DNA length. Instead, we expect that the increased intrinsic viscosity of the DNA polymer accounts for the slower coalescence. In theory, the viscosity of condensates should scale as a power of the molecular weight of the polymer (*Rubinstein and Colby, 2003*). However, under the solvent conditions tested here, and for DNA lengths < 3 kbp, the scaling relationship between intrinsic viscosity and DNA length is expected to be near linear, which has been confirmed experimentally (*Ross and Scruggs, 1968*; *Tsortos et al., 2011*). Thus, the increase in size of linear DNA from 147 bp to 2.7 kbp should approximately correspond to an order of magnitude change in viscosity. However, while coalescence was complete within 1 s for condensates formed with 147 bp DNA, condensates formed with 2.7 kbp DNA required several minutes to complete coalescence (*Figure 2D*). This greater than 100X increase in the rate of coalescence overshoots our expectations based solely on DNA length changes, demonstrating that HP1α-DNA interactions also contribute to the intrinsic viscosity of the condensate. Moreover, condensates formed with 9 kbp DNA (~60X larger than 147 bp) were unable to complete coalescence within an hour (*Figure 2D*). And while these condensates do exhibit a slow reduction in perimeter over time, suggesting that coalescence is proceeding locally, at the whole condensate level, the morphology of these condensates remains aspherical. Together these results indicate that within condensates, DNA is constrained by HP1α interaction networks leading to novel conformational restrictions and effective polymer interactions. Importantly, the length of heterochromatic domains in vivo is typically greater than 10 kbp. Therefore, the molecular interactions that occur in condensates formed around longer DNA molecules (9 kbp and longer) resulting in non-spherical morphologies may more closely mimic in vivo genomic environments.

Overall, these experiments suggest that HP1α and DNA differentially contribute to bulk droplet properties; the length of DNA and how it interconnects with HP1α interaction networks delimits condensate viscosity, while HP1α interactions likely define condensate surface tension. This means, that as the DNA length increases, the timescale for global conformational rearrangements of the DNA polymer also increases, while the timescale for rearrangements of HP1α-DNA and potentially HP1α-HP1α interactions are likely to remain fairly constant.

## HP1α dynamically binds to DNA while simultaneously maintaining stable DNA domains

To further investigate the interplay between these two types of rearrangements (HP1α-DNA and HP1α-HP1α vs. intra-DNA dynamics), we quantified the dynamics of HP1α and DNA within condensates We assessed the dynamics of HP1α using fluorescence recovery after photobleaching (FRAP). We find that for HP1α, despite large differences in droplet morphology, the rate of recovery is unaffected by changes in DNA length after partial photobleaching (*Figure 3A–C*). This result is consistent with HP1α-DNA and potential HP1α-HP1α interactions remaining unaffected by changes in DNA length. Condensates formed with DNA ranging in length from 147 bp to ~50 kbp showed recovery of fluorescence with comparable $t_{1/2}$ values (~2 s) (*Figure 3C*), which are strikingly similar to recovery rates of HP1α measured in vivo (*Cheutin et al., 2003*; *Festenstein et al., 2003*). Consistently, bleaching of the entire condensate also showed rapid recovery of fluorescence within experimental error of complete recovery (*Figure 3—figure supplement 1E*). These results demonstrate

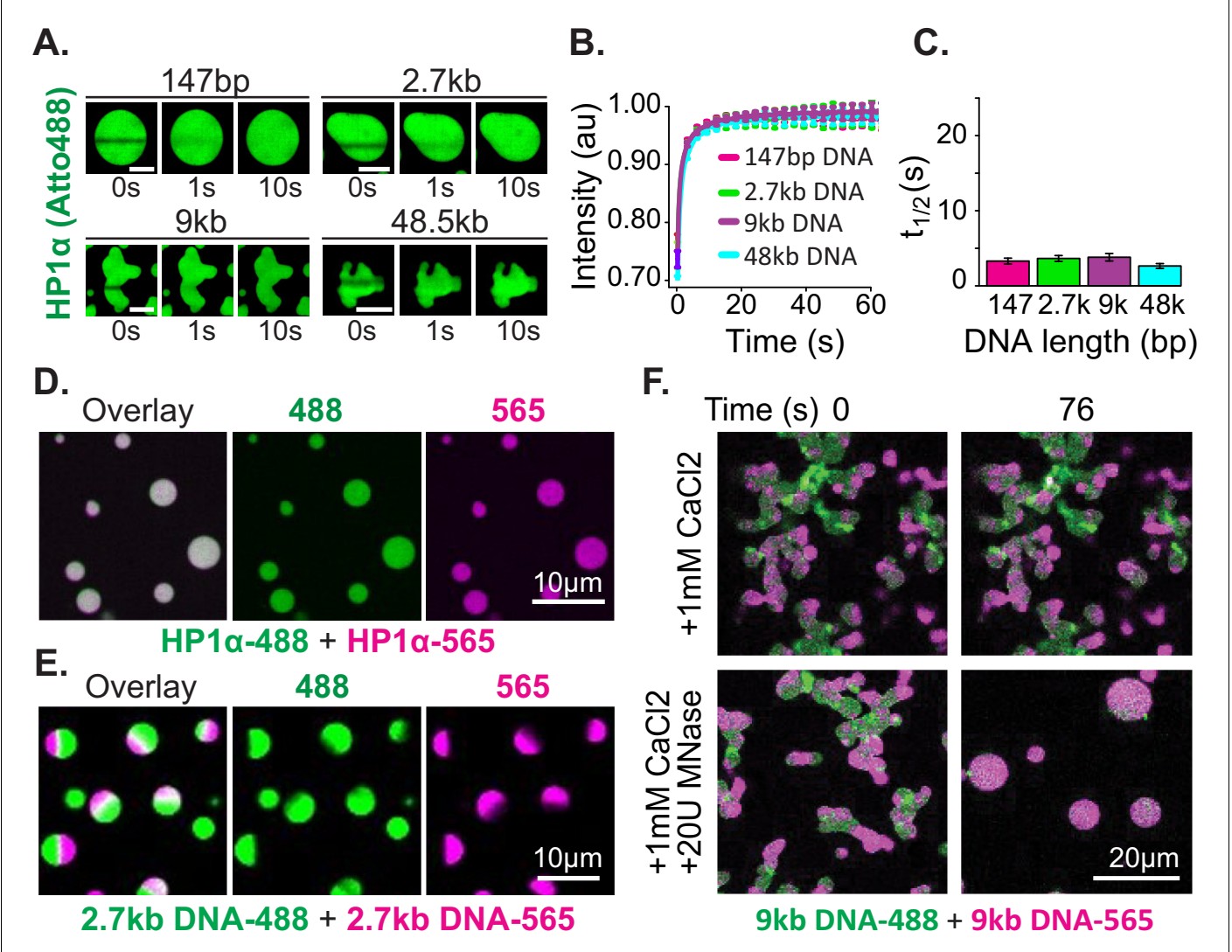

**Figure 3.** Distinct characteristics of HP1α and DNA in condensates. (A) FRAP of HP1α in condensates. Timestamped images from FRAP experiments for fluorescent HP1α and four lengths of linear DNA (147 bp, 2.7 kbp, 9 kbp, or 50 kbp). Scale bar indicates 5 μm. (B) Recovery of HP1α fluorescence intensity and (C) half-life of HP1α recovery plotted for each DNA length tested. N = 15 for each condition, error bars represent standard deviations. (D) Two-color HP1α mixing experiments. Condensates formed separately with 2.7 kbp unlabeled DNA and either HP1α−488 (green) or HP1α−565 (magenta) imaged 1.16 min after mixing. (E) Two-color DNA mixing experiments. Condensates formed separately with unlabeled HP1α and 2.7 kbp DNA-488 (green) or 2.7 kbp DNA-565 (magenta) imaged 4.4 min after mixing. (F) MNase treatment of condensates. Mixed condensates formed separately with unlabeled HP1α and 9 kbp DNA-488 (green) or 9 kbp DNA-565 (magenta) treated with either 1 mM CaCl₂ or 1 mM CaCl₂ and 20U MNase. Images shown for both conditions before and 76 s after the treatment.

The online version of this article includes the following figure supplement(s) for figure 3:

**Figure supplement 1.** Whole droplet FRAP of HP1α−488 in HP1α-DNA condensates.

**Figure supplement 2.** FRAP of DNA and mixing of HP1α and DNA in condensates.

that HP1α readily exchanges within condensates, and between condensate and solution populations, without disruption of the condensates. To further test the mobility of HP1α, we mixed pre-formed condensates prepared using HP1α labeled with either Atto488 (HP1α−488) or Atto565 (HP1α−565) (*Figure 3D*, *Figure 3—figure supplement 2D*). Within seconds after mixing, both HP1α−488 and HP1α−565 were found to have partitioned equally into all droplets (*Figure 3D*, *Figure 3—figure supplement 2D*). This rapid mixing of fluorescent protein is in full agreement with the FRAP estimates of HP1α mobility.

Next, we tested the mobility of the DNA polymer inside condensates. We performed mixing experiments using condensates preformed with HP1α and 2.7 kbp DNA that was end labeled with either Atto488 (DNA-488) or Atto565 (DNA-565) (*Figure 3E*, *Figure 3—figure supplement 2E*). The DNA length for these experiments was chosen to be long enough to manifest long polymer effects, but short enough to allow for the completion of coalescence (*Figure 2D–E*). Contrary to the observations above, DNA does not rapidly mix across condensates after fusion but is instead maintained in large and long lived (>1 hr) single-color sub-condensate domains (*Figure 3E*, *Figure 3—figure supplement 2E*). Furthermore, FRAP experiments of HP1α-DNA condensates labeled with YOYO-1 exhibit recovery rates proportional to DNA length: the longer the DNA, the slower the rate of recovery (*Figure 3—figure supplement 2A–C*).

These results confirm substantially different timescales for the mobility of HP1α versus DNA, as discussed in the previous section. Further, these results demonstrate that linear DNA as short as 3 kbp can be sustained in static compartments, despite prevalent and rapid exchange of HP1α. This outcome can arise through either the aforementioned viscosity and conformational constraints inherent to long DNA molecules, and/or through a collective activity of HP1α in condensates. To test if DNA viscosity is required for the persistence of sub-condensate DNA domains and non-spherical morphology, we dynamically altered the length of DNA in condensates by the addition of the calcium-dependent non-specific DNA nuclease, micrococcal nuclease. For these experiments, two-color HP1α-DNA condensates were formed using 9 kbp DNA resulting in diversely shaped condensates with alternating domains of fluorescence (*Figure 3F*). We expect that if polymer viscosity is required to maintain both the morphology of condensates and the reduced mobility of DNA, dynamically shortening the DNA length should result in both the resumption and completion of coalescence, and uniform mixing of fluorescent signals. Digestion of the DNA reveals this expectation to be accurate, and we observe rapid coalescence and mixing of alternately labeled DNA within condensates (*Figure 3F*). Importantly, we observe no effects on either phenomenon due to inclusion of calcium alone (*Figure 3F*).

Overall, these experiments reveal a remarkable character of HP1α-DNA condensates—a fast exchanging, liquid pool of HP1α can stably trap and organize large DNA molecules into isolated and long-lived domains. Seemingly, HP1α accomplishes this feat by increasing the effective viscosity of long DNA molecules to establish and maintain stable condensate structures. This rationale is consistent with our observation that changes to viscosity in HP1α-DNA condensates scale more sharply than expected from DNA length considerations alone. We note that the presence of nucleosomes will change the DNA length dependence of viscosity-driven effects. However, as we describe in the discussion, these differences will disappear at genomic scales and we expect that HP1 molecules will similarly increase the effective viscosity of chromatin to generate stable chromatin domains.

## HP1α maintains compacted DNA at relatively high forces

Given the dynamic behavior of HP1α, we expected that condensed HP1α-DNA structures, although kinetically long-lasting, would be readily dissolved if subjected to biologically relevant forces. To test this hypothesis, we investigated condensate stability against an externally applied force through optical trapping experiments combined with confocal microscopy (*Figure 4A–B*). In these experiments, we performed stretch-relax cycles (SRCs) (*Figure 4—figure supplement 1C*) by repetitively stretching and relaxing single DNA molecules in presence of HP1α. Simultaneously, we measured the force required to extend the DNA to a given length, yielding force-extension curves (*Figure 4C*, *Figure 4—figure supplement 1A*). Prior to adding HP1α, we first ensured that each tether was composed of a single molecule of DNA and behaved as previously described (*Figure 4C*; *Bustamante et al., 2000*). We then moved the trapped DNA molecule, held at an extension of ~5 µm, to a chamber containing HP1α and observed the formation of compacted HP1α-DNA structures analogous to those observed on DNA curtains (*Figures 1B* and *4B*). This initial incubation was sufficiently long to complete condensate formation (30 s). Notably, in this assay, compacted DNA structures appear in the center of the DNA molecule rather than at the end, because, with the motion of both ends of the DNA constrained by their attachment to polystyrene beads, the largest DNA chain fluctuations occur in the middle of the molecule.

For our initial experiments, DNA tethers bearing internal HP1α-DNA condensates were stretched at constant velocity to a final force of 40pN, immediately relaxed, and then stretched again (*Figure 4C*, *Figure 4—figure supplement 1A*). We observe a substantial deviation in the force

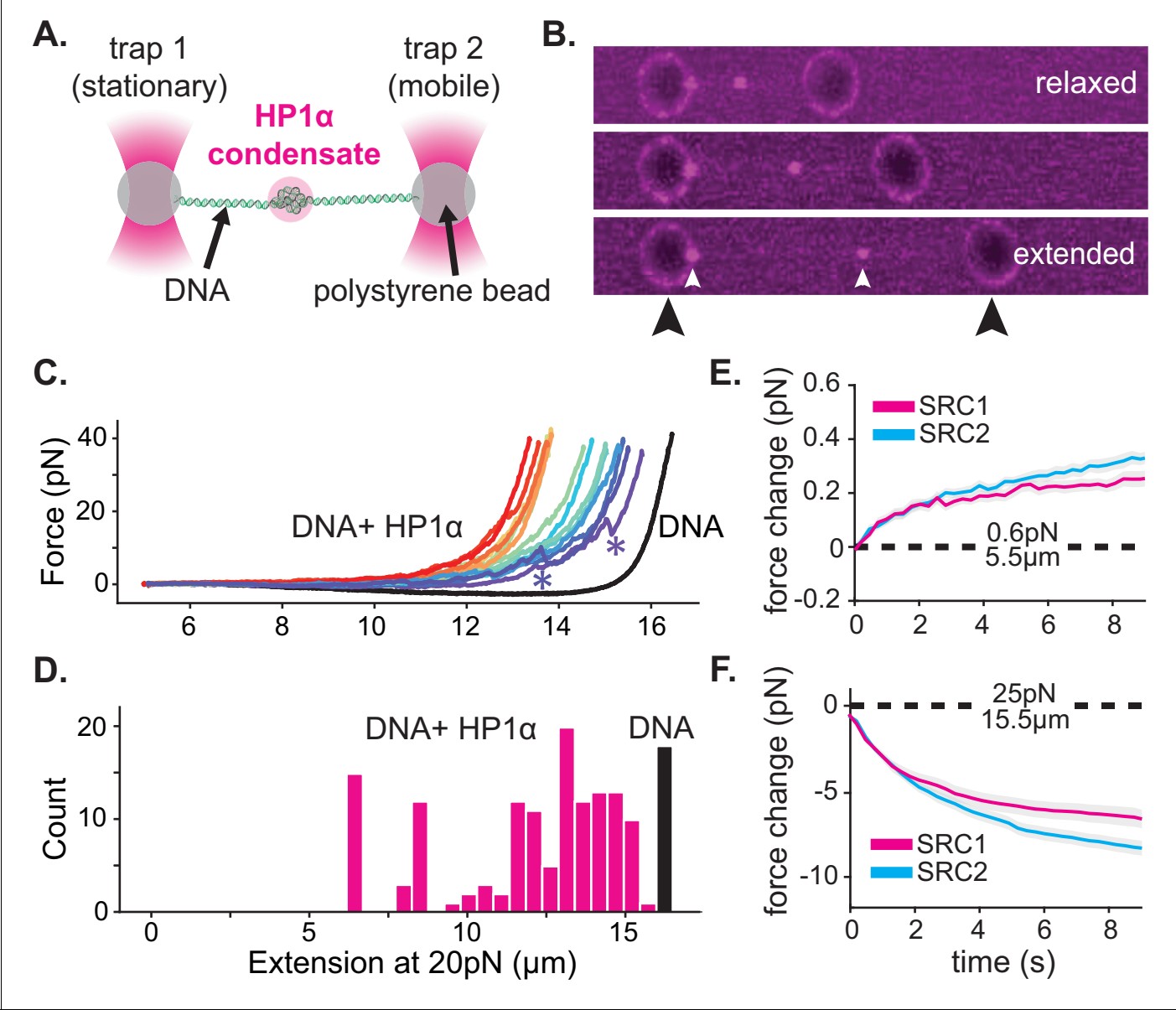

**Figure 4.** HP1α-DNA condensates resist disruptive forces and retain memory of past strain. (**A**) Cartoon of optical trap experiments. (**B**) Confocal images of relaxed, intermediate, and extended states of DNA (unlabeled) in the presence of HP1α (magenta). Black arrowheads indicate trapped beads and white arrowheads indicate HP1α-DNA condensates. (**C**) Force extension curves for DNA in the absence (black line) or presence of HP1α (colored lines). Each trace represents a single stretch-relax cycle (SRC) of the same DNA strand. Traces are colored by pulling order from first extension (violet) to the final extension (red). * indicates rupture event. (**D**) Histogram of DNA extension at 20 pN in the absence (black) or presence of HP1α (magenta). N = 150, 10 DNA strands pulled fifteen individual times each. (**E** and **F**) Force change for DNA incubated with HP1α in (**E**) relaxed or (**F**) extended conformation. Shown is the average of the first (magenta) and second (cyan) SRC. Data are averaged over 17 DNA strands, gray shaded region represents SEM.

The online version of this article includes the following figure supplement(s) for figure 4:

**Figure supplement 1.** Representative traces and controls for optical trap experiments.

extension curve for DNA in the presence of HP1α relative to DNA alone (*Figure 4C*, *Figure 4—figure supplement 1A*). We verified that the shift to larger forces for DNA extended in the presence of HP1α is not a consequence of radiation driven cross-linking (*Figure 4—figure supplement 1B*). From this measurement, we identify three prominent features of HP1α-DNA interactions. First, sequestered DNA domains, measuring on average 10 kbp, are able to resist disruption to an

instantaneous force of 40pN (*Figure 4C–D*). However, smaller HP1α-DNA structures (~1–2 kbps) are observed to rupture at lower forces ranging from 5 to 20pN, suggesting the stability of HP1α-compacted DNA scales by size (*Figure 4C*, *Figure 4—figure supplement 1A*, see '*'). Second, by integrating the area between the force-extension curves for DNA alone and in the presence of HP1α, we estimate that an average energetic barrier of ~1 $k_b$T/bp of compacted DNA separates HP1α-compacted states of DNA from extended DNA states in the absence of HP1α (*Figure 4C*, *Figure 4—figure supplement 1A*). Finally, we observed that each successive SRC resulted in more DNA stably sequestered by HP1α (*Figure 4C*). This surprising result shows that, after HP1α-DNA condensates are subjected to strain, polymer rearrangements and/or force-dependent selection of HP1α-binding interactions provide a basis for further stable compaction of DNA by HP1α.

Next, we asked whether or not HP1α-DNA condensates could compact DNA against force or maintain the compacted state when subjected to sustained force by performing consecutive SRCs that included waiting periods after complete relaxation (~5.5 μm) and after stretching to 25pN (~15.5 μm) (*Figure 4E–F Figure 4—figure supplement 1C–E*). During the waiting period after relaxation, we observe a steady force increase over time (*Figure 4E*, *Figure 4—figure supplement 1D–E*). This result may be the product of either association of HP1α molecules from solution and/or rearrangements of DNA and already bound HP1α. To test whether low-force DNA compaction required a constant influx of HP1α binding, we moved the DNA tether from the chamber containing HP1α to a chamber containing only buffer and performed an additional three SRCs (*Figure 4—figure supplement 1D*). We find that even in the absence of free HP1α, the population of already bound HP1α molecules is sufficient to induce compaction in the low force regime (~1 pN) (*Figure 4—figure supplement 1D*). Notably, compaction in the absence of free protein can be abrogated by increasing the ionic strength of the buffer (from 70 mM to 0.5M KCl) (*Figure 4—figure supplement 1E*), consistent with salt-induced decompaction observed on DNA curtains (*Larson et al., 2017*).

When the DNA is held at a steady extension of 15.5 μm following stretching, we observe a drop in measured force over time (*Figure 4F*, *Figure 4—figure supplement 1D–E*). This relaxation indicates that HP1α-DNA condensates are biased toward disassembly during sustained higher forces. This result is potentially due to force-dependent changes in the kinetics of HP1α binding and/or the reduction in DNA strand fluctuations required by HP1α to induce compaction. To test whether HP1α in solution could affect the stability of the condensate, through a facilitated exchange mechanism (*Graham et al., 2011*), we again performed an additional three SRCs in the absence free HP1α (*Figure 4—figure supplement 1D–E*). We find that the disassembly of HP1α-DNA condensates at higher forces proceeds at the same rate irrespective of the presence of HP1α in solution (*Figure 4—figure supplement 1D*).

Notably, during both waiting periods—before and after stretching—we measure changes in HP1α-DNA condensation activity in later SRCs (*Figure 4E–F*, *Figure 4—figure supplement 1D–E*). In the relaxed configuration, during low-force compaction, we observe more robust compaction during the second SRC relative to the first (*Figure 4E*). In comparison, we observe more rapid disassembly while waiting at higher forces during the second SRC (*Figure 4F*). These strain-induced effects on HP1α behavior can have important consequences for how HP1α-organized genetic material responds to cellular forces *Amy et al., 2020*. For example, RNA polymerase ceases to elongate when working against forces as low as 7.5–15pN (*Galburt et al., 2007*). Our experiments show that short transient bursts by polymerase are unlikely to disassemble and may even strengthen HP1α-compacted structures above the force threshold for efficient transcription. However, repeated, sustained efforts by polymerase might be sufficient to relax HP1α-compacted structures and allow for transcription to proceed.

Moreover, these data suggest that a dynamic network of HP1α-DNA and potential HP1α-HP1α interactions can account for both increased viscosity and stabilization of global condensate structure. In general, we propose that such properties arise from a mean-field activity of an exchanging population of HP1α molecules that constrain the DNA at any given time. That is, regardless of the stability of any individual HP1α molecule, the average character of the HP1α-DNA network is maintained in condensates at a pseudo steady state.

While the measured stability of HP1α-DNA condensates is consistent with a role for HP1α as a mediator of transcriptional repression, it is hard to reconcile this activity with dynamic chromatin reorganization when cellular cues necessitate the disassembly of heterochromatin. These data also raise the question of which molecular features of HP1α allow it to realize its many functions in

condensates and on single DNA fibers. Below we first study the molecular features of HP1α that drive condensate formation and then address how HP1α-DNA condensates may be disassembled.

### The hinge domain of HP1α is necessary and sufficient for DNA compaction and condensate formation

First, we set out to determine the smallest piece of HP1α sufficient for the collective HP1α behaviors on DNA we have observed. HP1α is comprised of three disordered regions interspaced by two globular domains: a chromodomain (CD) and a chromoshadow domain (CSD) (*Figure 5A*; *Canzio et al., 2014*). The CD binds to di- and tri-methylation of lysine 9 on histone 3 (H3K9me) and the CSD mediates HP1 dimerization as well as interactions with other nuclear proteins (*Canzio et al., 2011*; *Eissenberg et al., 1990*; *Kaustov et al., 2011*; *Smothers and Henikoff, 2000*). The central disordered region, or hinge domain, of HP1α mediates DNA binding (*Meehan et al., 2003*; *Smothers and Henikoff, 2001*). Finally, the N-terminal extension (NTE) and the C-terminal extension

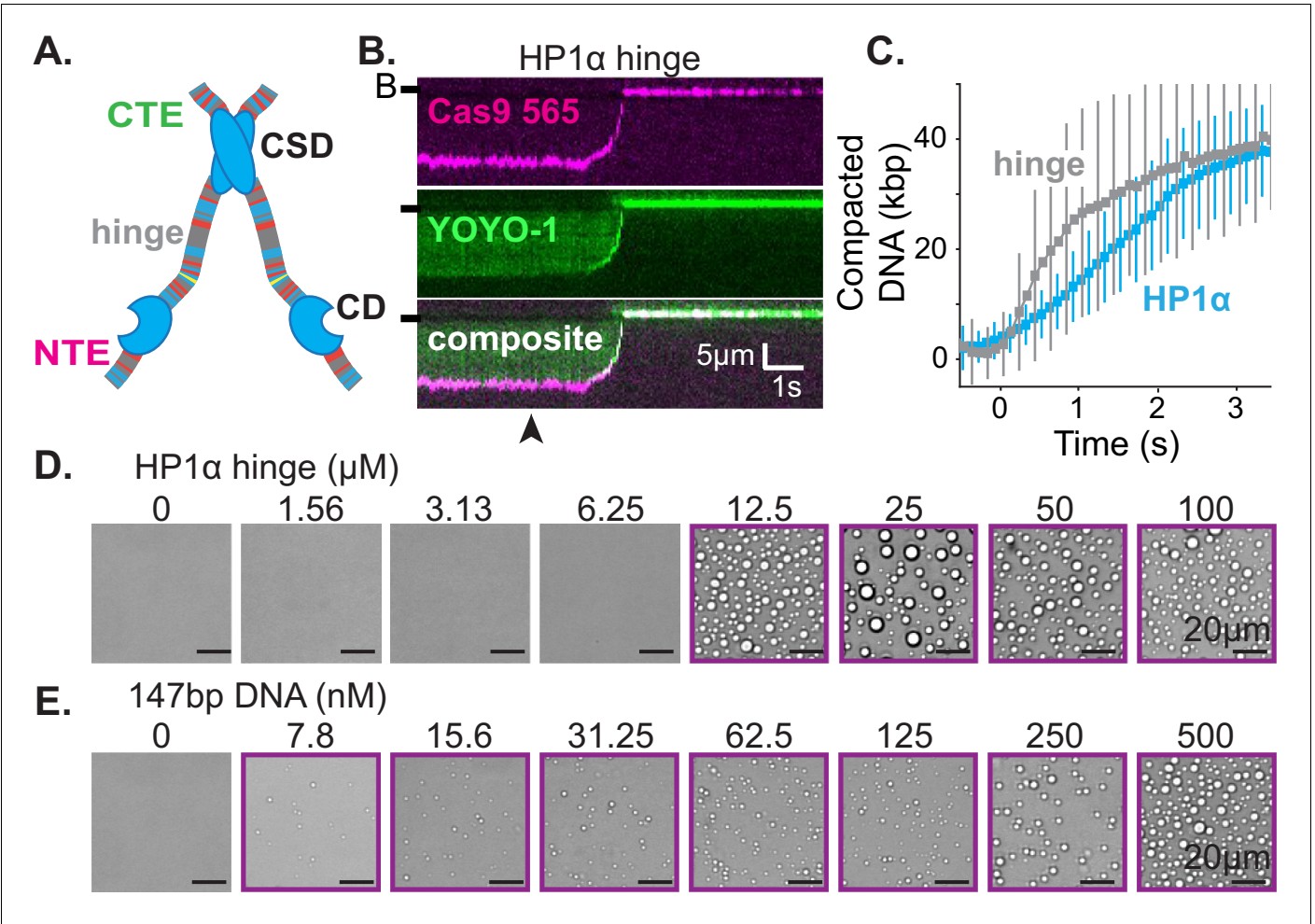

**Figure 5.** The hinge region of HP1α is sufficient for DNA compaction and condensate formation. (A) Cartoon of HP1α with color-coded disordered regions: positive residues (K and R) blue, negative residues (E and D) red, proline yellow, and all other residues gray. Key HP1α domains are labeled: chromodomain (CD), chromoshadow domain (CSD), hinge, N-terminal extension (NTE), and C-terminal extension (CTE). (B) Kymogram of DNA compaction by the hinge domain. DNA is labeled with dCas9 (top) and YOYO-1 (middle), also shown as composite image (bottom). Arrowhead represent estimated time of protein injection. (B-) or (-) specifies location of the barrier. (C) Average DNA compaction by 5 µM HP1α (N = 157) and 5 µM HP1α-hinge (N = 169), error bars represent standard deviations. (D and E) Bright-field images of the HP1α-hinge and DNA. (D) Titration of the HP1α-hinge with 500 nM 147 bp DNA. (E) Titration of 147 bp DNA with 12.5 µM HP1α-hinge. Purple boxes indicate presence of condensates. The online version of this article includes the following figure supplement(s) for figure 5:

**Figure supplement 1.** The hinge region of HP1 is sufficient for DNA compaction.

(CTE) of HP1α have been shown to regulate oligomerization of phosphorylated HP1α (*Larson et al., 2017*). While all five domains of HP1α collaborate to determine in vivo localization, cellular localization at heterochromatic sites is completely abolished by mutations to the hinge domain of HP1α (*Cheutin et al., 2003*; *Kellum et al., 1995*; *Yuan and O'Farrell, 2016*). Therefore, we first investigated the activity of the hinge domain isolated from the rest of the protein. Surprisingly, not only is the hinge domain sufficient for DNA compaction (*Figure 5B–C*, *Figure 5—figure supplement 1B–C*), but also compaction proceeds at twice speed of the full-length protein (*Figure 5C*, *Figure 1—figure supplement 1C*, *Figure 5—figure supplement 1C*). Additionally, the hinge domain is sufficient to induce the formation of condensates with DNA (*Figure 5D–E*). And, even with short (147 bp) DNA oligomers, the critical concentration for condensate formation is reduced by a factor of four relative to full-length HP1α (from 50 to 12.5 μM) (*Figures 2A* and *5D*). Surprisingly, the critical concentration is reduced, and DNA compaction increased, even though the valency of the hinge domain alone is ostensibly half that of full-length HP1α due to removal of the CSD. Furthermore, and consistent with observations of full-length HP1α, condensates formed with the hinge domain exhibit a continuous reduction in size upon lowering DNA concentration, rather than exhibiting a sharp transition between the presence and absence of droplets (*Figures 2A* and *5E*). The strong in vitro activity of the hinge domain alone compared to full-length HP1α, and the requirement of an unperturbed hinge domain for proper function in vivo, raise the possibility that the remaining disordered regions of HP1α exist to regulate the behavior of the hinge domain.

## The disordered extensions of HP1α regulate hinge domain activity

Previous work demonstrated that the NTE and CTE of HP1α play opposing roles in controlling the phase-separation behavior of phosphorylated HP1α (*Larson et al., 2017*). In this context, the CTE acts in an auto-inhibitory role and phosphorylated residues in the NTE promote oligomerization through interactions with the hinge domains in trans (*Figure 6—figure supplement 1A*). We hypothesized that these two disordered terminal extensions may similarly regulate hinge domain activity in the context of DNA-driven HP1α phase-separation. To test this possibility, we deleted these extensions of HP1α, either separately or in tandem (*Figure 6—figure supplement 2A*).

Removal of the NTE (HP1α-ΔNTE) abolished detectable condensate formation with short DNA oligomers and increased the critical concentration for condensate formation with longer DNA (*Figure 6A–B*). Furthermore, HP1α-ΔNTE compacted DNA ~20 times slower than full-length HP1α and only managed to compact ~7 kbp of the available ~50 kbps (*Figure 6C,E*, *Figure 6—figure supplement 2B,C*). These results suggest that removal of the NTE lowers the apparent on-rate for DNA binding, and generally raises the free energy of HP1α-DNA interactions. However, the compacted structures that do form in our curtain assay persist even after the pulse of HP1α-ΔNTE protein exits the flowcell, suggesting that removing the NTE of HP1α might not compromise the off-rate of HP1α (*Figure 6—figure supplement 2C*). The inhibition of both DNA compaction and condensate formation upon NTE deletion demonstrates that the NTE plays a positive role in each process. Furthermore, these effects are also consistent with, but not definitive of, the NTE contributing to higher order oligomerization of HP1α in the context of DNA binding (see below).

In contrast, deletion of the CTE (HP1α-ΔCTE) decreased the critical concentration for condensation with 147 bp DNA oligomers an order of magnitude (*Figure 6A–B*). This result indicates that removal of the CTE lowers the free energy of HP1α-DNA condensation. HP1α-ΔCTE also compacted DNA three times faster than full-length HP1α and almost twice the apparent rate measured for the hinge domain alone (*Figure 6C–D*, *Figure 5—figure supplement 1C*, *Figure 6—figure supplement 2C*). Together with the compaction activity of the hinge and HP1α-ΔCTE, these data demonstrate that the CTE negatively regulates the activity of the hinge domain in the context of full-length HP1α. This is consistent with previous crosslinking mass-spectrometry studies that indicate the CTE binds to the hinge when not bound to DNA (*Larson et al., 2017*).

Finally, when both the NTE and CTE are removed from HP1α (HP1α-ΔNTEΔCTE), we observe intermediate phenotypes: compaction rates faster than HP1α-ΔNTE but slower than HP1α-WT, HP1α-ΔCTE, or the hinge alone (*Figure 6C–E*, *Figure 5—figure supplement 1C*, *Figure 6—figure supplement 2C*) and a decrease in the critical concentration for HP1α-DNA condensation, although not to the same extent as HP1α-ΔCTE (*Figure 6A–B*). This result further supports our model of opposing regulation of the hinge domain by the NTE and CTE of HP1α in the context of the full-length protein.

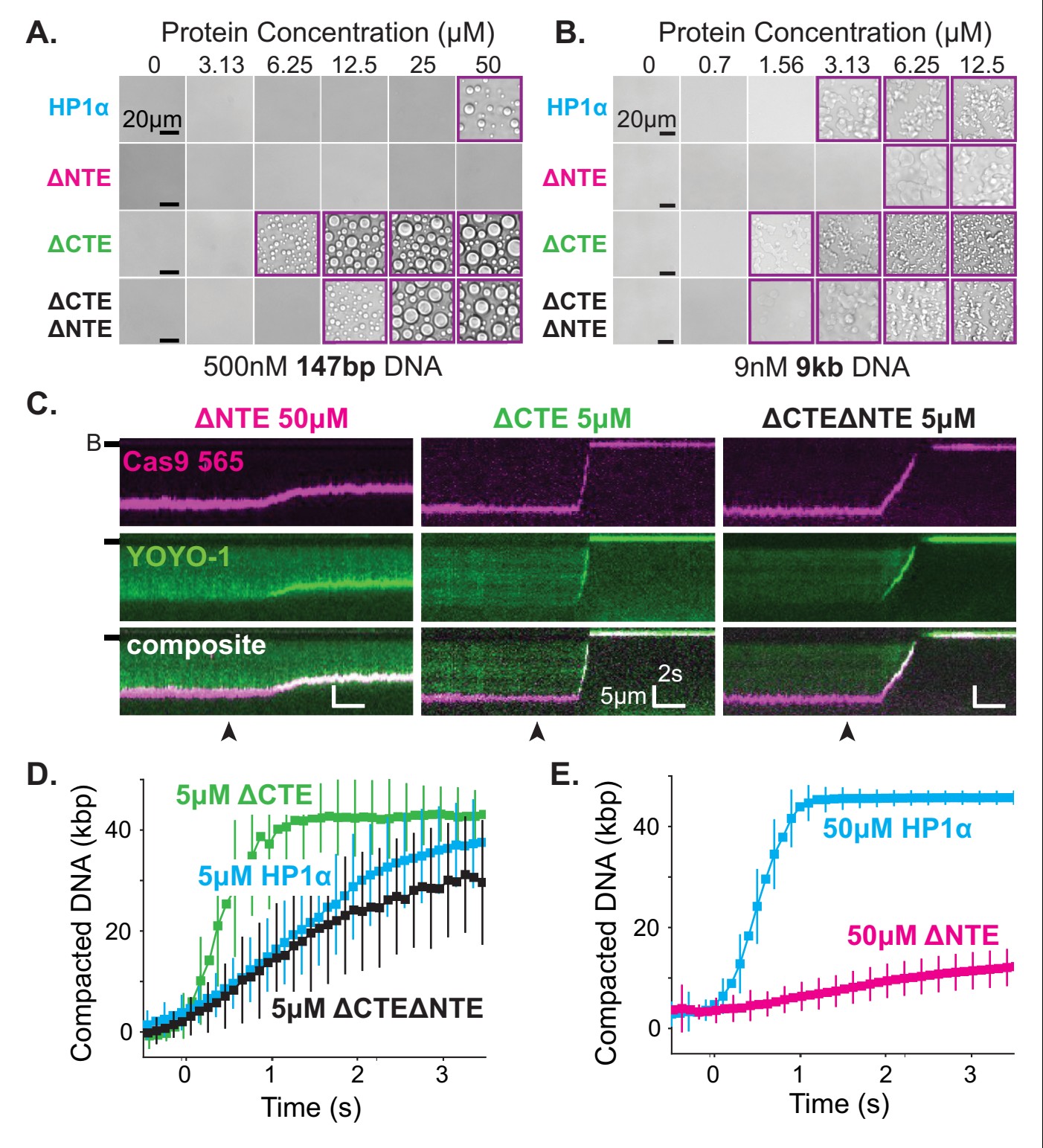

**Figure 6.** The disordered extensions of HP1α regulate DNA compaction and condensate formation. (A and B) Bright-field images of HP1α domain mutants and DNA. (A) Titration of HP1α domain mutants with 500 nM 147 bp DNA. (B) Titration of HP1α domain mutants with 9 nM 9 kbp DNA. Purple boxes indicate presence of condensates. (C) Kymograms of DNA compaction by HP1α domain mutants. DNA is labeled with dCas9 (top) and YOYO-1 (middle), also shown as composite image (bottom). Data shown for reactions including 50 μM HP1αΔNTE, 5 μM HP1αΔCTE, and 5 μM HP1αΔNTEΔCTE, respectively. Arrowheads represent estimated time of protein injection. (B-) or (-) specifies location of the barrier. (D) Average DNA

*Figure 6 continued on next page*

*Figure 6 continued*

compaction by 5 μM HP1α (N = 157), 5 μM HP1αΔCTE (N = 96), and 5 μM HP1αΔCTEΔNTE (N = 89). (E) Average DNA compaction by 50 μM HP1α (N = 272) and 50 μM HP1αΔNTE (N = 163). In (D) and (E) error bars represent standard deviations.

The online version of this article includes the following figure supplement(s) for figure 6:

**Figure supplement 1.** Proposed model of HP1α autoregulation and potential oligomerization.

**Figure supplement 2.** DNA compaction activity of HP1α domain mutants.

The findings above reveal that the HP1α hinge is sufficient for condensate formation with DNA and that its activity is regulated by the CTE and NTE of HP1α. In previous sections, we have shown that full-length HP1α binds to DNA and induces local compaction that nucleates and supports the growth of phase separated domains. Now it is clear that these behaviors are subject to, and resultant of, a complex and coordinated network of interactions between the domains of HP1α (*Figure 6—figure supplement 1A*). This regulation of activity likely occurs between the disordered domains of individual HP1α molecules and also across many HP1α molecules throughout HP1α-DNA complexes.

## Differential droplet formation and DNA compaction by HP1 paralogs

The three human HP1 paralogs, differ significantly in sequence across their unstructured regions (*Figure 7A–B*; *Canzio et al., 2014*). Our results thus far suggest these differences should manifest differential activities with DNA and offer a convenient approach to study the regulation of HP1α's hinge domain by the NTE and CTE. First, we tested each paralog's ability to compact DNA (*Figure 7C–E*). We find that HP1β displays a substantially reduced rate of DNA compaction relative to HP1α (*Figure 7C,E*). This indicates a relative deficiency in the apparent interaction strength between HP1β and DNA. Indeed HP1β's compaction activity is more comparable to that of HP1α-ΔNTE (*Figure 6—figure supplement 2C*, *Figure 7—figure supplement 1A–C*). Yet, despite slower compaction, HP1β continues to sustain compacted DNA even after the bulk of the injected pulse of HP1β has passed through the flowcell (*Figure 7E*, *Figure 7—figure supplement 1A–C*). This suggests a lower bound for HP1β's off-rate from compacted DNA on the order of minutes. In comparison, HP1γ compacts DNA more rapidly than HP1β, although HP1γ does not achieve the rapid compaction rates of HP1α (*Figure 7D–E*, *Figure 7—figure supplement 1A–C*). Moreover, HP1γ rapidly disassembles as the concentration of free HP1γ in the flowcell begins to decline (*Figure 7E*, *Figure 7—figure supplement 1A–C*). We propose that this effect is the result of HP1γ having a faster off-rate from DNA relative to HP1α or HP1β. Importantly, these experiments suggest that genomic regions organized by HP1α and HP1β would require less protein for maintenance and be more resistant to disruption relative to domains organized by HP1γ.

Next, we tested our interpretation of compaction experiments by assessing the relative capacity of each HP1 paralog to drive condensate formation with DNA (*Figure 7F–G*). We predicted that due to its decreased compaction rate, HP1β would struggle to form condensates with DNA. However, if any condensates form, we would predict that those HP1β-DNA structures would be stable. On the contrary, we expect HP1γ will readily condense into liquid domains with DNA but require a higher concentration to maintain condensation relative to HP1α, due to the apparent increase in reversibility of compaction on curtains (*Figure 7D–E*, *Figure 7—figure supplement 1A–C*). We find that HP1γ does form condensates with 3 kbp DNA, although the critical HP1γ concentration required to induce droplet formation is, in fact, higher than that for HP1α (*Figure 2E*, *Figure 7F*). Moreover, HP1γ does not form condensates with 147 bp DNA, under conditions where HP1α continues to drive DNA condensation (*Figure 7G*). These results are consistent with a lower DNA-binding affinity and higher off-rate for HP1γ. In contrast, we find that HP1β does not induce droplet formation regardless of the length of co-incubated DNA (*Figure 7G*, *Figure 7—figure supplement 1D*). This result mirrors the attenuated condensate forming activity of HP1α-ΔNTE and is consistent with lower DNA compaction rates and a lower DNA binding affinity. Furthermore, HP1β demonstrates that the ability to induce and maintain stable DNA compaction in it of itself is not definitive of condensate formation.

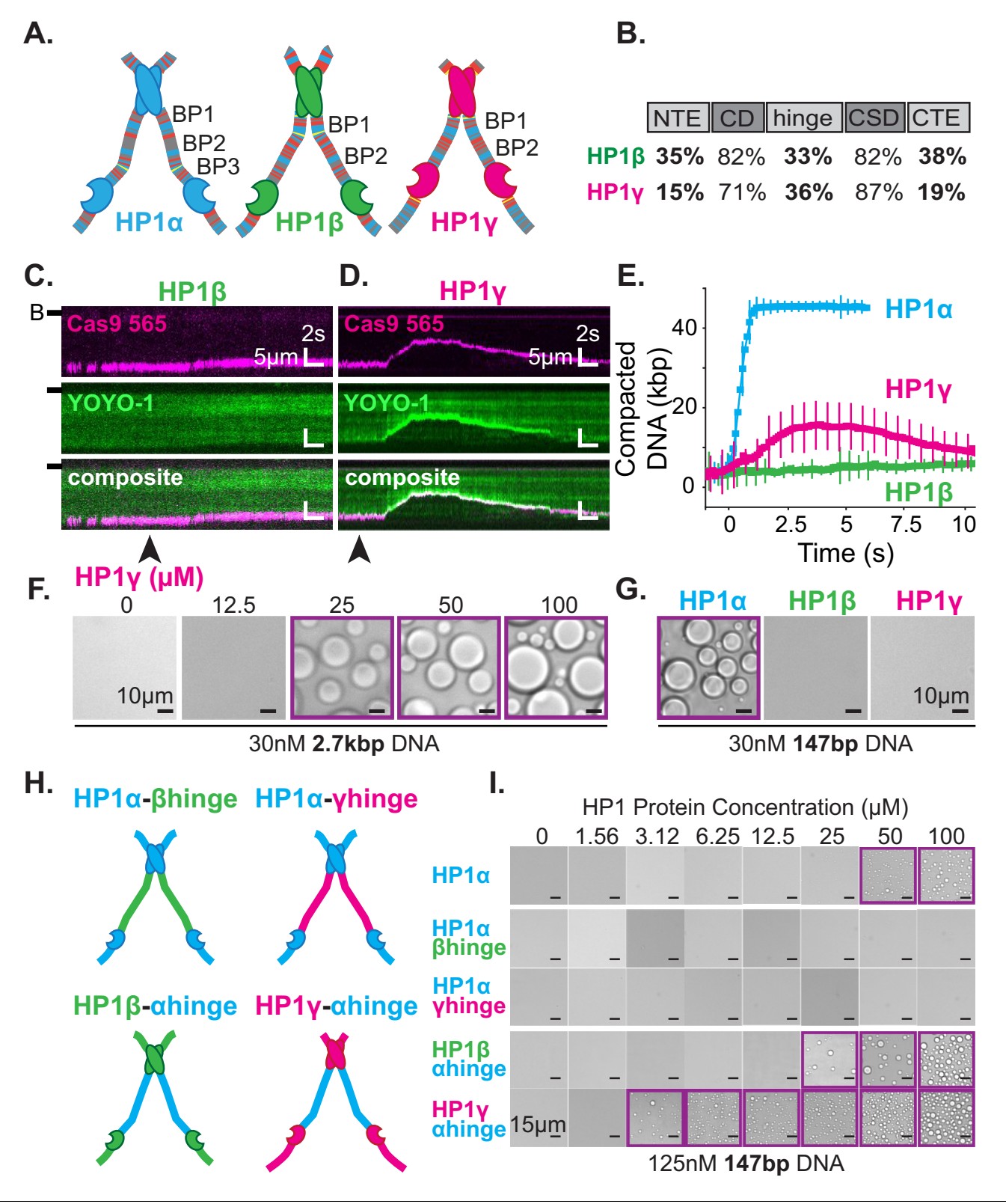

**Figure 7.** DNA compaction and condensate formation activity of HP1β and HP1γ. (**A**) Cartoons of the three paralogs of human HP1 with color-coded disordered residues: positive residues (**K and R**) blue, negative residues (**E and D**) red, proline yellow, and all other residues gray. Basic patches (BP) for each paralog are labeled. (**B**) Comparison of amino acid homology between HP1α and HP1β or HP1γ. (**C and D**) Kymograms of DNA compaction by (**C**) HP1β and (**D**) HP1γ. DNA is labeled with dCas9 (top) and YOYO-1 (middle), also shown as composite image (bottom). Arrowheads represent estimated

*Figure 7 continued on next page*

*Figure 7 continued*

time of protein injection. (B-) or (-) specifies location of the barrier. (E) Average DNA compaction by 50 μM HP1α (N = 272), HP1β (N = 86), and HP1γ (N = 54). Error bars represent standard deviations. (F) Bright-field images of HP1γ and 2.7 kbp DNA. (G) Bright-field images of 100 μM HP1α, HP1β, or HP1γ and 147 bp DNA. (H) Cartoon of HP1 hinge domain swaps. (I) Bright-field images of HP1 domain swap mutants and 147 bp DNA. Purple boxes indicate presence of condensates.

The online version of this article includes the following figure supplement(s) for figure 7:

**Figure supplement 1.** DNA compaction by HP1β and HP1γ.

## The disordered regions of HP1 paralogs drive differential DNA compaction and condensate formation activity

The above results uncovered substantial differences in the abilities of HP1β and HP1γ to compact and form condensates with DNA as compared to HP1α. We presumed these differences in activity are due to differences in their respective disordered domains. Specifically, we expect the disparities across paralogs in their hinge domain, which for HP1α is sufficient for DNA compaction and condensation (*Figure 5B–E*), to be the strongest predictor of activity. To directly determine the differences in activity due to individual hinge domains, we replaced the hinge domain of HP1α with the corresponding hinge domains from either HP1β or HP1γ, respectively (HP1α-βhinge and HP1α-γhinge) (*Figure 7H*). We find that both mutants fail to produce condensates in the presence of DNA (*Figure 7I*), demonstrating that, within the context of full-length HP1α and the HP1 paralogs, the HP1α's hinge domain is necessary for droplet formation. While it might have been expected for HP1α-γhinge to exhibit some condensate formation activity, it is worth noting that HP1γ lacks any appreciable CTE, and its NTE is remarkably different than that of HP1α (*Figure 7A–B*). Therefore, in its native context, the hinge domain of HP1γ likely does not have to navigate autoregulation in order to promote productive HP1γ-DNA interactions.

We then performed compensatory swaps of the hinge domain of HP1α into HP1β (HP1β-αhinge) and HP1γ (HP1γ-αhinge) (*Figure 7H*). We find both these mutants now readily form condensates with DNA, demonstrating the HP1α hinge is also sufficient for phase separation in the context of the other HP1 paralogs (*Figure 7I*). Intriguingly, the critical concentration for condensate formation was decidedly lower for both α-hinge mutants than for HP1α; two-times lower for HP1β-αhinge and ten-times lower for HP1γ-αhinge (*Figure 7I*). These results indicate that the HP1α hinge is more active outside of its native context where it is free from the inhibitory effect of its CTE.

The HP1 paralogs are often found in overlapping genomic regions in cells. Given the differential activities of the paralogs, we next asked now mixed populations might manifest distinct properties in condensates by performing droplet assays in the presence of paralog competitors. When HP1β or HP1γ were premixed with HP1α and added to DNA to assess condensate formation, both HP1β and HP1γ inhibited droplet formation in a concentration-dependent fashion (*Figure 8A*). Notably, these experiments were performed with 147 bp DNA, which when incubated with HP1γ, does not induce condensate formation (*Figure 7G*). Interestingly, when introduced to pre-formed HP1α-DNA condensates, HP1β is capable of invading and subsequently dissolving condensates at a rate proportional to HP1α exchange (*Figure 8B*). In contrast, HP1γ does not destabilize, but rather enriches in the pre-formed HP1α-DNA condensates (*Figure 8C*). These results may simply be a reflection of binding site competition. However, the HP1 paralogs have been suggested to heterodimerize, so it is attractive to hypothesize that heterodimers between HP1α and HP1β or HP1γ have lower DNA-binding affinity or disrupted regulatory interactions such that condensate formation is inhibited. Furthermore, while it is difficult to account for the differences in pre-formed condensate disruption by HP1β and HP1γ with a simple steric occlusion model, differences in heterodimerization activity and/or activity of heterodimers provide an acceptable rationale.

Together, these results suggest inter-paralog competition as a possible mechanism of cellular regulation of HP1-mediated chromatin domains. Moreover, these experiments demonstrate the critical advantage of biological organization by liquid condensates—competition can be fast. Fast competition means that, regardless of domain stability, when the molecular environment changes, condensates can respond to those changes at the rate at which the organizing material exchanges. For condensation of DNA by HP1α, this means that even in the context of highly viscous, tangled DNA and large networks of protein-protein and protein-DNA interactions that resist mechanical

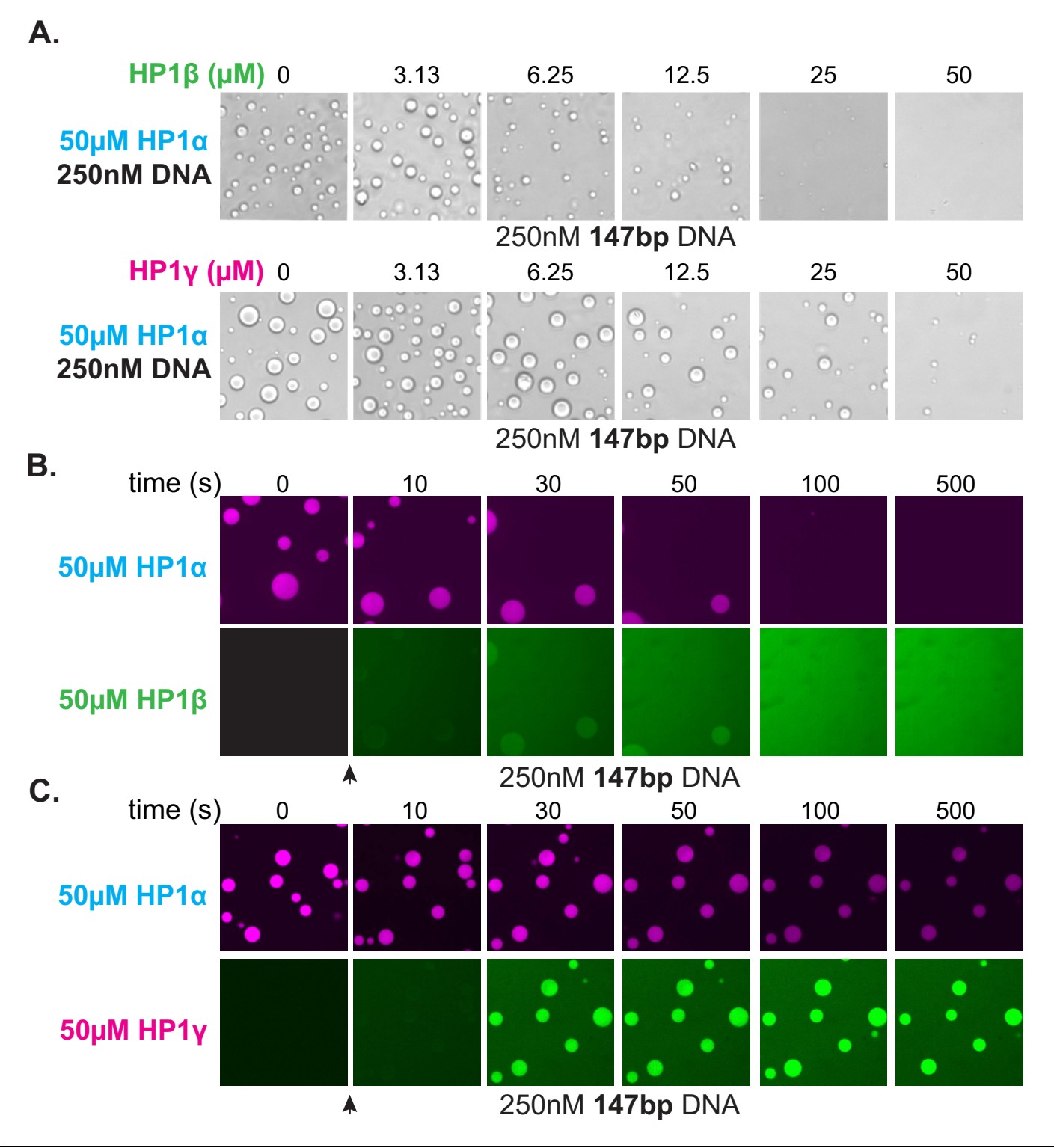

**Figure 8.** Effect of HP1β and HP1γ on HP1α-DNA condensate formation and stability. (A) Bright-field images of DNA and pre-incubated mixtures of HP1α and HP1β (top) or HP1α and HP1γ (bottom). (B and C) Confocal images showing a time course of HP1α condensates after injection of (B) HP1β or (C) HP1γ.

disruption at steady state, domains can easily be disassembled in seconds due to the rapid exchange rate of individual HP1α molecules.

## Discussion

Heterochromatin serves to organize large regions of the eukaryotic genome into domains that are positionally stable yet can be disassembled in response to cell cycle and developmental cues (*Cheutin et al., 2003*; *Cheutin and Cavalli, 2012*; *Dion and Gasser, 2013*; *Gerlich et al., 2003*; *Kind et al., 2013*; *Marshall et al., 1997*). Previous work on HP1-mediated heterochromatin uncovered several key biophysical properties of HP1 proteins such as the ability to form oligomers and to form liquid-like phase-separated condensates with DNA and chromatin (*Canzio et al., 2011*; *Kilic et al., 2018*; *Larson et al., 2017*; *Sanulli et al., 2019*; *Strom et al., 2017*; *Wang et al., 2019*). A closer examination of these properties can help discern their cellular influence and ultimate role in regulation of heterochromatin states in cells. However, a major challenge in such an endeavor has been connecting the actions of individual HP1 molecules on DNA to the collective phenotype of a heterochromatin domain. Here, we have used a series of complementary assays that allow us to measure the mesoscale behavior of human HP1 proteins and interpret that behavior in terms of single molecule activity. Our findings indicate at least three regulatable steps by which HP1α organizes and compacts DNA (*Figure 9A–C*): (1) Local assembly of HP1α along DNA prior to DNA condensation; (2) initiation of DNA compaction through capturing of proximal DNA fluctuations via HP1α-DNA and HP1α-HP1α interactions to form a proto-condensate, and (3) progression of DNA compaction through inclusion of uncompacted DNA into the growing condensate via HP1α-DNA and HP1α-HP1α interactions. We further find that the polymer behavior of DNA, together with the ability of HP1α molecules to make multivalent interactions with rapid on/off kinetics, results in stable mesoscale structures that resist mechanical forces but are subject to competition (*Figure 9C–D*). Finally, comparison of the behavior of HP1α with that of HP1β and HP1γ uncovers new biophysical differences between the three paralogs. Below we discuss the mechanistic and biological implications of our findings in the context of previous observations.

### Implications for regulation of heterochromatin assembly and spreading

The framework presented above has implications for understanding how heterochromatin domains grow through incorporation of additional regions of the genome. Specifically, factors that lower the affinity of HP1α for DNA, or potentially HP1α's affinity for itself, will result in reduced formation of compacted DNA and a heightened sensitivity to disruption. Regions of DNA that are low-affinity binding sites for HP1α will also resist incorporation into compacted domains and can potentially act as insulating sites against HP1α activity.

Furthermore, in our experiments, we find that longer DNA promotes the formation of HP1-DNA condensates (*Figures 2A,E*, *6A–B* and *7F–G*). This observation is consistent with longer DNA, with higher valency, increasing the local concentration of proto-condensates. Therefore, restricting the continuity of HP1α binding sites in vivo would also be predicted to inhibit growth of heterochromatin domains. An obvious way to interrupt continuous stretches of DNA is by the presence of nucleosomes. Indeed >70% of mammalian genomes are occupied by nucleosomes (*Chereji et al., 2019*). The traditional view is that H3K9me3 containing nucleosomes act as platform for HP1 interactions that impart preference for heterochromatin versus euchromatin (*Lachner et al., 2001*). In this context, it is tempting to speculate that histone modifications act to restore HP1-binding sites interrupted by the nucleosome core, thereby promoting HP1 assembly and specificity. At the same time, the presence of nucleosomes would also regulate the architecture of HP1 assembly. Consistent with such a possibility, HP1 proteins from *S. pombe* have been shown to bridge across and deform H3K9me3 nucleosomes (*Sanulli et al., 2019*).

Interactions made by HP1 proteins with the histone octamer and H3 tail may serve additional roles in regulating the stability of the condensate. Under the ionic conditions that more closely approximate those within the nucleus, our results suggest that the critical concentration of HP1α for forming HP1α-DNA condensates is higher than the total HP1α concentrations estimated in certain cell types (*Figure 2—figure supplements 2*, 4) (*Larson et al., 2017*; *Müller et al., 2009*). We propose that HP1α-interactions that increase affinity, like binding to H3K9me chromatin through the

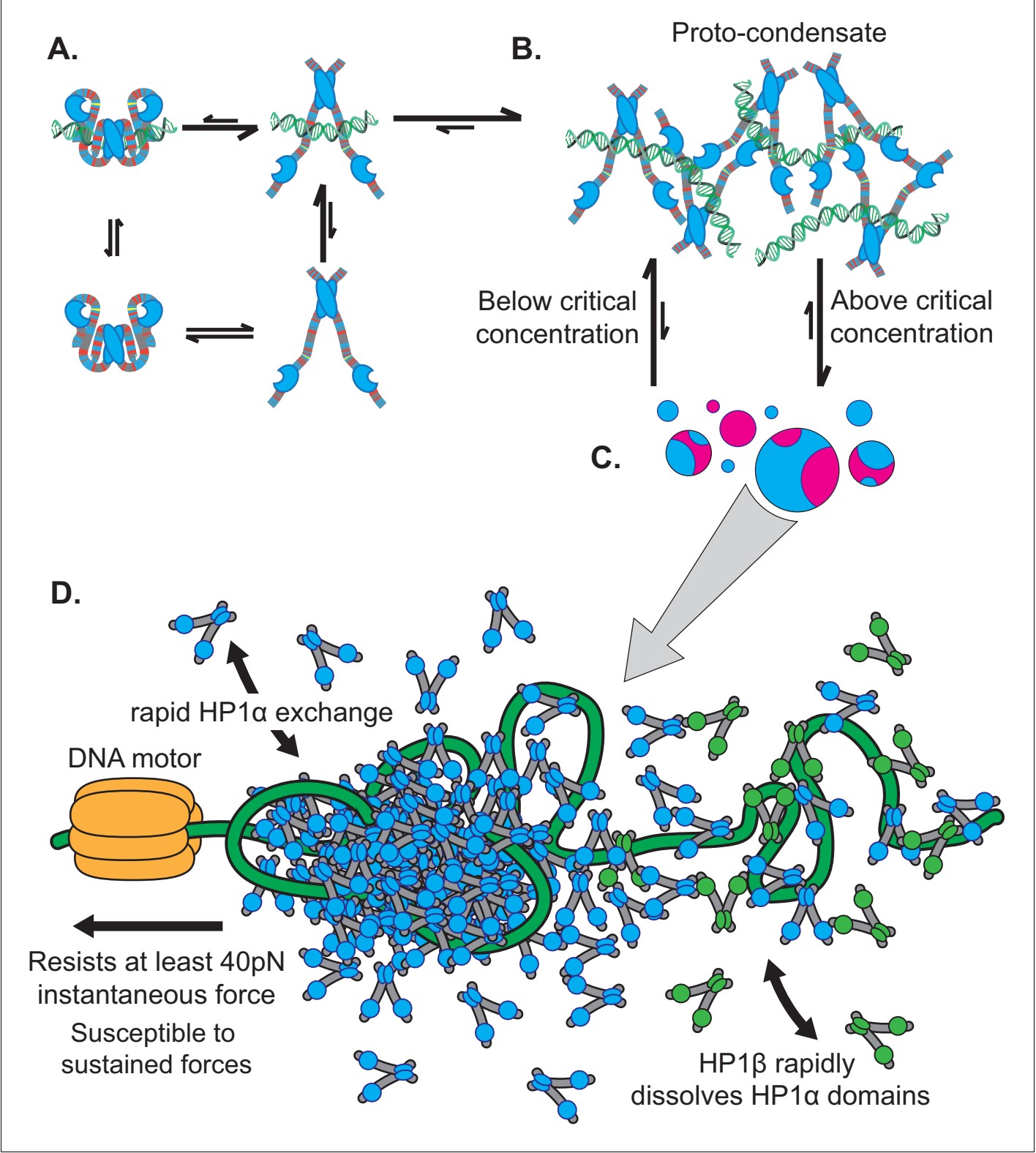

**Figure 9.** Microscopic to macroscopic activity of HP1α. (**A**) At the microscopic scale, interactions between the terminal extensions and hinge domain toggles HP1α between autoinhibited and active states. DNA biases HP1α to the active state. (**B**) At the intermediate scale, HP1α and DNA cluster into proto-condensates. (**C**) If HP1α is present above the critical concentration, proto-condensates aggregate into large macroscopic droplets characterized by liquid behavior of HP1α and static DNA held in sub-condensate domains. (**D**) At genomic loci, HP1α condensates are remodeled by forces, resisting

*Figure 9 continued on next page*

*Figure 9 continued*

and strengthening in response to instantaneous forces, but relaxing and weakening in response to sustained forces. HP1α domains are also subject to disruption and reinforcement from HP1-interacting proteins like HP1β.

chromodomain can serve to increase the local HP1α concentration over the critical value for condensate formation.

The internal regulatory network of interactions across the hinge, NTE, and CTE regions of HP1α will also influence assembly on chromatin in vivo. Our results imply that protein binding or post-translational modification of the NTE and CTE could regulate the local concentration of HP1α, and thereby the ability of HP1α to condense chromatin. For example, proteins that bind to the CTE may induce HP1α to behave more like HP1α-ΔCTE promoting condensation (*Figure 6A–D*, *Figure 6—figure supplement 2B-C*). A large number of nuclear proteins bind HP1α in close proximity to the CTE, including two proteins shown to modulate HP1α phase separation in vitro, Lamin-B receptor and Shugoshin (*Larson et al., 2017*; *Smothers and Henikoff, 2000*). Alternatively, modifications may provide the basis for new interactions as seen when the N-terminal of HP1α is phosphorylated (*Larson et al., 2017*).

Importantly, because HP1α concentrations in the cell are similar to the lower limit for condensate formation that we observe in vitro (*Figure 2E*), the assembly of HP1α is well-poised to be influenced by molecular interactions and modifications (*Müller et al., 2009*).

## Implications for the versatility of heterochromatin function

A major function of heterochromatin is the compartmentalization of the genome (*Allshire and Madhani, 2018*). In this context, our results indicate a dominant role for the DNA polymer in regulating its own compartmentalization. In condensates, we find that ~3 kbp pieces of DNA are fixed in place on the order of an hour, while HP1α molecules can diffuse on the order of seconds (*Figure 3*, *Figure 3—figure supplement 1*, *Figure 3—figure supplement 2*). Our results imply that such behavior arises from two sources: the intrinsic viscosity of DNA due to its polymer properties, and the mean activity of rapidly rearranging HP1α molecules, which creates an average protein-DNA network equivalent to a set of static interactions. As a result, when two condensates fuse, the HP1α molecules rapidly exchange between the two condensates while the DNA from each condensate remains trapped in separate territories (*Figure 3D–E*, *Figure 3—figure supplement 2D–E*).

Inclusion of nucleosomes substantially increases the persistence length and linear density of DNA while potentially decreasing the number of HP1-binding sites (*Bystricky et al., 2004*). Since HP1 interactions also contribute to viscosity, any effects from a reduction in HP1-binding sites would be balanced by the increased rigidity of the chromatin polymer. Additionally, in the context of chromatin, HP1 molecules can use additional domains, such as the CD and the CSD, to further constrain chromatin through interactions with H3K9me modifications and the histone core, respectively. In all these considerations it is important to note that the length effects due to the large sizes of chromatin domains in the nucleus would overshadow differences between the viscosities of chromatin versus DNA. Thus, we propose that the meso-scale behaviors observed in the context of DNA will be recapitulated in the context of chromatin, but with additional regulatable steps.

From a charge passivation perspective, the ability of HP1α to condense DNA bears similarities to counterion mediated condensation of DNA by ions such as spermidine (*Bloomfield, 1997*). Interestingly, spermidine-mediated DNA condensates dissolve upon application of ~1 pN force requiring only ~0.1 kT/bp of work in contrast to the >1 kbT/bp required to disassemble HP1α-DNA condensates (*Figure 4C*; *Baumann et al., 2000*). Some of these differences may arise from the specific DNA-binding properties of the hinge region as opposed to those of spermidine. However, at 40 mM KCl, HP1α can phase-separate in the absence of DNA indicating an intrinsic ability for self-association (*Figure 2—figure supplement 2C*). We therefore propose that the stabilization of compacted DNA achieved by HP1α may arise from its ability to form HP1α-HP1α interactions in addition to HP1α-DNA interactions.

Importantly, we find that HP1α-DNA condensates are able to resist disruption by instantaneous forces of at least 40pN (*Figure 4C*, *Figure 4—figure supplement 1A*). Furthermore, we find that transient forces increase the ability of condensates to resist subsequent disruptions (*Figure 4C,E*,

*Figure 4—figure supplement 1D–E*). The high resistance to force, as well as the conversion to a more stable state upon application of transient external force, provides a biophysical explanation for how heterochromatin can confer mechanical stability in two contexts: to the nucleus when the nuclear membrane is subjected to mechano-chemical signaling events, and to centromeres when they are subjected to forces of chromosome segregation (*Allshire and Madhani, 2018*; *Amy et al., 2020 Stephens et al., 2019*). However, we also show that sustained high forces provoke the relaxation of condensates and sensitize condensates for subsequent disruptions (*Figure 4F*, *Figure 4—figure supplement 1D–E*). These effects highlight the ability of HP1-mediated heterochromatin to be shaped by cellular forces.

Our results further explain how mechanically stable and long-lived domains are dissolved in response to cellular cues. We find that, even while the global character of HP1-DNA condensates is fixed, the constituent HP1 molecules are highly dynamic (*Figure 3*, *Figure 3—figure supplements 1* and *2*; *Kilic et al., 2018*). This dynamism allows for rapid competition and interference, and, because the organizing network of HP1-DNA interactions is built from weak transient encounters, results in swift disassembly of structures and dispersal of material (*Figure 8B*). More generally, because condensates often rely on the integrated weak interactions of large populations to build cellular structures, they also present low energetic barriers to competition. Condensates thus present unique advantages in the context of cellular organization. It is, however, worth noting that competition need not be direct and the chemical environment in condensates can also restrict competitor access to internal structures. The general organizational principles that we have uncovered here can be applied in many biological contexts but seem most readily applicable to the unique functions and constraints shared by genome organizing proteins.

## Implications for paralogs and evolution

In addition to HP1α, there are two other paralogs of HP1 in humans: HP1β, which in some cell types is suggested to be less abundant than HP1α, and HP1γ which may exist at similar levels to HP1α, again in certain cell types (*Bártová et al., 2005*). Despite sharing similar domain architecture and conservation of sequence, these paralogs of HP1 differentially localize in the cell and perform individual functions (*Figure 7A–B*; *Minc et al., 1999*; *Nielsen et al., 2001*). Importantly, each paralog also performs distinctly in our two assays. We find that HP1β binds to DNA at a lower rate than HP1α, leading to reduced DNA compaction activity; yet, the compaction by HP1β is relatively stable (*Figure 7C,E*, *Figure 2—figure supplement 2D*, *Figure 7—figure supplement 1A–C*). Additionally, we find that HP1β is unable to produce observable condensates with DNA (*Figure 7G*). This may be because HP1β is deficient in modes of DNA binding, is unable to engage in protein-protein interactions beyond dimerization, its central disordered region is ill adapted to condensation, or any combination therein. Notably, when HP1β also makes nucleosomal contacts, it can compact chromatin leading to condensation (*Hiragami-Hamada et al., 2016*; *Wang et al., 2019*). Additionally, HP1β is a particularly effective competitor for HP1α in vitro, suggesting that HP1β interactions may be adapted for tempering HP1α-organized chromatin or to serve a role in establishing chromatin boundaries (*Figure 8A–B*).

Furthermore, we find HP1γ binds to DNA at a much faster rate than HP1β, but HP1γ-DNA condensates also rapidly disassemble in the absence of excess free protein, resulting in rapid compaction followed by rapid decompaction on DNA curtains (*Figure 7D–E*, *Figure 2—figure supplement 2D*, *Figure 7—figure supplement 1A–C*). Yet, HP1γ is able to induce condensate formation with DNA, though at a higher protein concentration than HP1α (*Figure 7F–G*). Notably, in certain cells, HP1γ is the most diffuse HP1 paralog, often not exhibiting localization at all, which might be the result of the high instability we observe (*Minc et al., 1999*; *Nielsen et al., 2001*). The higher critical concentration for HP1γ-DNA condensation reflects a higher setpoint for regulation in comparison to HP1α, meaning HP1γ will require a larger cellular investment in protein levels to induce condensation. It is also possible that higher order chromatin organization by HP1γ may be at cross purposes with the known role of HP1γ in promoting transcription elongation (*Vakoc et al., 2005*).

The three human HP1 paralogs are the result of past gene duplications, and while they have faithfully conserved their chromo- and chromoshadow domains, their disordered regions have diverged completely (*Figure 7B*; *Levine et al., 2012*; *Lomberk et al., 2006*). It is possible that each paralog achieves specificity in biological function through their disordered regions. We demonstrate this possibility by exchanging disordered domains among the paralogs, converting HP1β and HP1γ into

robust agents of DNA condensation (*Figure 7H–I*). These experiments also reveal the evolutionary potential of the modular HP1 domain architecture. For example, it is easy to imagine the effect that inserting a sequence with variable condensing ability into HP1α would have on the genome and heterochromatin stability. Indeed, the molecular diversity of HP1 proteins across eukaryotes suggests that evolution has already taken advantage of HP1 architecture (*Lomberk et al., 2006*).

## Implications for the diversity of biological liquid-like phase-separation phenomenon

In addition to heterochromatin, phase-separation phenomena have been observed in many different biological contexts, including the nucleolus, P bodies, and P granules (*Banani et al., 2017*; *Elbaum-Garfinkle et al., 2015*; *Feric et al., 2016*; *Lin et al., 2015*; *Smith et al., 2016*). Recent studies have also found evidence for phase-separation behavior in the context of transcription and DNA repair (*Cho et al., 2018*; *Kilic et al., 2019*). The increasing number of observations of phase-separation in biological systems has created an apparent need to define the criteria for liquid phase-separated condensates (*McSwiggen et al., 2019b*). Some commonly proposed criteria for liquid-like phase-separation behavior are, (i) A boundary that confines the mobility of phase-separating molecules, (ii) concentration buffering, and (iii) differential viscosities inside vs outside of condensates (*Banani et al., 2017*; *Erdel et al., 2020*; *McSwiggen et al., 2019b*). Some of these criteria are based on assumptions of a homogenous solute (condensed phase) and solvent (surrounding phase). However in vivo, both the solute and solvent are heterogenous. Below we discuss how this, and other considerations make the above criteria limiting in the context of biologically meaningful condensates.

i. Boundaries: It has been proposed that phase-separated condensates will have boundaries that promote preferential movement of phase-separating molecules inside the condensate as opposed to movement of molecules across the boundary. To measure such a property, recent studies have assessed the permeability of condensate boundaries to the entry and exit of GFP-HP1α molecules using FRAP, and shown that entry of GFP-HP1α molecules from the outside of the condensate occurs at least as fast as internal mixing of GFP-HP1α molecules (*Erdel et al., 2020*). However, for rapidly moving molecules, like HP1α, and small domain sizes, like chromocenters, differences in recovery rates due to internal mixing versus exchange across the boundary will be near resolution limits. Furthermore, in vitro, some liquid condensates have been demonstrated to have surprisingly low density and high permeability (*Wei et al., 2017*). In these condensates, it is possible to decouple the movement of molecules within condensates from their mesoscale droplet properties. Indeed, our results show that HP1α molecules in vitro can mix both within and across condensates at comparable rates (*Figure 3A*, *Figure 3—figure supplement 2A–C*, *Figure 3—figure supplement 1*). Furthermore, our measured rates of HP1α exchange (seconds) in condensates are likely to make measurements of differential dynamics within versus without condensates difficult to distinguish. Importantly, the results from our FRAP studies of HP1α-DNA condensates in vitro reveal exchange rates that are similar to prior FRAP studies on heterochromatin puncta in cells (*Cheutin et al., 2003*; *Erdel et al., 2020*; *Festenstein et al., 2003*). Overall, our results demonstrate that some categories of condensates can display mesoscale liquid-like characteristics even while the motion of molecules within condensates and across their boundaries are similar.

ii. Concentration buffering: Concentration buffering refers to a phenomenon where increasing the total concentration of a condensing protein, like HP1α, does not change the concentration inside relative to outside of condensates (*Vicsek and Family, 1984*). Instead, the volume of condensates increases. In opposition, recent studies have shown that increasing the cellular concentrations of HP1α by overexpression does not result in an increase in the size of heterochromatin puncta but instead increases the concentration of HP1α inside the puncta (*Erdel et al., 2020*). Interestingly, our in vitro data is consistent with some expectations of concentration buffering of HP1α above the critical concentration. Specifically, we show that the size of HP1α-DNA condensates grows with the addition of either DNA or HP1α (*Figure 2C*). However, it is important to note that partitioning of material into condensed and soluble phases is also defined by the energetics of the molecular interactions in either compartment. In the in vitro context, concentration buffering in HP1α-DNA condensates would be achieved if there is only competition for binding interactions between HP1α molecules, and if the chemical environment inside of the condensates does not vary as a function

of this competition. However, it is also possible that additional HP1-HP1 interactions may change the concentration buffering behavior of HP1α-DNA condensates. Further, in cells, heterochromatin is expected to contain other DNA-binding factors in addition to HP1α. Therefore, it is possible that increasing concentrations of HP1α simply compete off other component molecules within heterochromatin puncta. Critically, it has recently been shown that this assumption of concentration buffering also fails to describe the concentration dependence of protein inclusion in the nucleolus, perhaps the best-defined phase-separated organelle in the cell (*Riback et al., 2020*).

iii. Differential viscosities: In vitro studies using nucleolar components have shown that some nucleolar proteins such as NPM1 substantially increase the bulk viscosity of condensates compared to water (*Feric et al., 2016*). This result has led others to define liquid-like phase-separated condensates by whether or not they exhibit increased viscosity relative to the outside dilute phase (*Erdel et al., 2020*). However, there are limitations to using differential viscosities as a defining feature of liquid-like phase-separation. For example, a recent study used fluorescence correlation microscopy of GFP-HP1α to measure the viscosity of GFP-HP1α in chromocenters in cells. Based on similar rotational diffusion of GFP-HP1α inside and outside of heterochromatin puncta, the study concluded that the HP1α in chromocenters does not experience a higher viscosity relative to elsewhere in the nucleus and therefore chromocenters do not conform to the definitions of LLPS (*Erdel et al., 2020*). However, as opposed to micro-rheology experiments which measure bulk viscosities (*Feric et al., 2016*), rotational diffusion reports on a convoluted energetic landscape defined by a multitude of interactions made by the diffusing molecules. Specifically, in this case, the conclusion that HP1α does not engage in LLPS in vivo relies on the validity of the assumption that there is a distinct separation between the strength and abundance of molecular interactions inside relative to outside condensates. For simple in vitro systems diffusion of condensing material will almost always be slower inside condensates rather than outside, as the outside represents a dilute molecular environment. However, in the nucleus, there is no analog of the 'dilute phase' as the majority of the nucleus is crowded with diverse molecules. Thus, it is possible for HP1α to make interactions both within and outside of condensed heterochromatin in vivo that results in comparably slowed mobilities relative to a dilute solution.

The discussion above about the complexity of the cellular environment relative to in vitro experiments raises the general question of how in vitro demonstrations of liquid-like phase-separation can be used to derive biologically meaningful insights. At a foundational level, quantitative in vitro experiments are essential to detail the properties of biological condensates, as we have done for the HP1-proteins. The complexities of cellular contexts can then be layered on, and tested, in a systematic manner. Critically, determination that the simplest assumptions of LLPS behavior derived from in vitro studies are not upheld in vivo is extremely valuable in identifying the effects of such additional complexity (*Erdel et al., 2020*). But, if molecules exhibit liquid-like phase separation activity in vitro, the interactions that produce that behavior do not vanish in the cell. Rather they are integrated into the complex network of cellular interactions that spans all of the molecules in the cell. And importantly, the interactions that give rise to macroscopic LLPS in vitro are present even among sparingly few molecules, regardless of whether they manifest across scales into large liquid domains. Sometimes those interactions will be obscured by cellular activity, but in other contexts those same interactions may be at the forefront biological activity. In vitro studies are therefore essential to provide a framework to test and interpret the relative behaviors of phase-separating components in cells.

Overall, our findings here underscore that, as new activities of biological condensates continue to be discovered it is important to characterize the biophysical nature of these condensates and the biologically relevant properties that they enable.

# Materials and methods

## Protein purification
### General method
Rosetta competent cells (Millipore Sigma 70954) transformed with expression vectors for 6x-HIS tagged HP1 proteins (*Supplementary file 1*) were grown at 37°C to an OD600 of 1.0–1.4 in 1 L of 2xLB supplemented with 25 μg/mL chloramphenicol and 50 μg/mL carbenicillin. HP1 protein

expression was induced by the addition of 0.3 mM isopropy-βD-thiogalactopyranoside (IPTG). Cells were then grown for an additional 3 hr at 37C, before pelleting at 4000xg for 30 min. Cell pellets were then resuspended in 30 mL Lysis Buffer (20 mM HEPES pH7.5, 300 mM NaCl, 10% glycerol, 7.5 mM Imidazole) supplemented with protease inhibitors (1 mM phenylmethanesulfonyl fluoride (Millipore Sigma 78830), 1 μg/mL pepstatin A (Millipore Sigma P5318), 2 μg/mL aprotinin (Millipore Sigma A1153), and 3 μg/mL leupeptin (Millipore Sigma L2884)). Cells were then lysed using a C3 Emulsiflex (ATA Scientific). Lysate was clarified by centrifugation at 25,000xg for 30 min. The supernatant was then added to 1 mL of Talon cobalt resin (Takara 635652) and incubated with rotation for 1 hr at 4℃. The resin-lysate mixture was then added to a gravity column and washed with 50 mL of Lysis Buffer. Protein was then eluted in 10 mL of elution buffer (20 mM HEPES pH 7.5, 150 mM KCl, 400 mM Imidazole). Then, TEV protease was added to cleave off the 6x-HIS tag and the protein mixture was dialyzed overnight in TEV cleavage buffer (20 mM HEPES pH 7.5, 150 mM KCl, 3 mM DTT) at 4℃. The cleaved protein was then further purified by isoelectric focusing using a Mono-Q 4.6/100 PE column (GE Healthcare discontinued) and eluted by salt gradient from 150 mM to 800 mM KCl over 16 column volumes in buffer containing 20 mM HEPES pH 7.5 and 1 mM DTT. Protein containing fractions were collected and concentrated in a 10K spin concentrator (Amicon Z740171) to 500 μL and then loaded onto a Superdex-75 Increase (GE Healthcare 29148721) sizing column in size exclusion chromatography (SEC) buffer (20 mM HEPES pH7.5, 200 mM KCl, 1 mM DTT, 10% glycerol). Protein containing fractions were again collected and concentrated to 500 μM in a 10K spin concentrator. Finally, aliquots were flash frozen in liquid nitrogen and stored at −80℃.

HP1α, HP1β, and HP1γ were all purified as described above. For the terminal extension deletes (HP1αΔNTE, HP1αΔCTE, and HP1αΔNTEΔCTE) (*Supplementary file 1*), minor changes to the ionic strength of buffers were made. Specifically, each protein was dialyzed into a low-salt TEV protease buffer (20 mM HEPES pH 7.5, 75 mM KCl, and 3 mM DTT) in the overnight cleavage step. Additionally, the salt gradient used in isoelectric focusing ranged from 75 mM to 800 mM KCl. The rest of the protocol followed as written above.

The HP1α hinge was purified as written until the overnight TEV cleavage step. After which, the protein was loaded onto a Hi-Trap SP HP column (GE Healthcare 17115201) and eluted in a salt gradient from 150 mM to 800 mM KCl over 16 column volumes in buffer containing 20 mM HEPES and 1 mM DTT. Protein containing fractions were collected and concentrated in a 10K spin concentrator to 500 μL and then loaded onto a Superdex-30 10/300 increase (GE Healthcare 29219757) sizing column in size exclusion chromatography (SEC) buffer. Protein containing fractions were then collected and concentrated to 500 μM in a 10K spin concentrator. Finally, aliquots were flash frozen in liquid nitrogen and stored at −80℃.

## Protein labeling

Proteins constructs for fluorescent labelling were modified to contain a C-terminal GSKCK tag and to substitute native reactive cysteines to serine residues (HP1α-C133S and HP1γ-C176S) (*Supplementary file 1*). For labeling, HP1 proteins were dialyzed overnight into SEC buffer with 1 mM TCEP substituted for DTT. Protein was then mixed at a 1:1 molar ratio with either maleimide Atto488 or maleimide Atto565 (Millipore Sigma 28562, 18507). The reaction was immediately quenched after mixing by addition of 10x molar excess of 2-mercaptoethanol. Labeled protein was then separated from free dye over a Hi-Trap desalting column in SEC buffer (GE Healthcare 17-1408-01). Labeled protein was then flash frozen in liquid nitrogen and stored at −80℃.

## DNA purification

Plasmids containing DNA used in this study were amplified in DH5α cells (ThermoFisher 18265017) grown in TB. Plasmids were purified using a Qiagen Plasmid Giga kit (Qiagen 12191) Plasmids containing the '601' DNA sequence were digested with EcoRV (NEB R0195S) and the 147 bp fragments were then isolated from the plasmid backbone by PAGE purification. Briefly, DNA were loaded into a 6% acrylamide gel and run at 100 mV for ~2 hr in 1xTBE. The desired 147 bp DNA band was cut out of the gel and soaked in TE (10 mM Tris-HCL pH 7.5, 1 mM ETDA) buffer overnight. The supernatant was then filtered, and DNA isolated by two sequential ethanol precipitations. The 2.7 kbp DNA (Puc19) was linearized by HindIII (NEB R0104S) digestion and purified by two sequential

ethanol precipitations. The 9 kbp DNA (pBH4-SNF2h [*Leonard and Narlikar, 2015*]) was linearized by BamHI (NEB R0136S) digestion and purified by two sequential ethanol precipitations.

DNA from bacteriophage λ (λ-DNA) (NEB N3011S) used in phasing and curtains experiments was prepared by heating to 60°C to release base pairing of the cohesive ends in the presence of complementary 12 bp primers as previously described (*Greene et al., 2010*). For curtain experiments, the primer targeted to the 3′ overhang of λ-DNA was modified to include a 5′ biotin. λ-DNA and primers were then allowed to slowly cool to room temperature and then incubated overnight with T4 DNA ligase (NEB M0202S). The λ-DNA was then precipitated in 30% PEG(MW 8000) + 10 mM MgCl$_2$ to remove excess primers and washed three times in 70% ethanol before resuspension and storage in TE.

### DNA labeling

DNA was end-labeled with fluorescent dUTPs as follows. A total of 50 μg linear 2.7 kbp and 9 kbp plasmids were incubated with 12.5 units of Klenow 3′→ 5′ exo– (NEB M0212S), 33 μM dATP, dCTP, dGTP (Allstar scientific 471-5DN), and either 33 μM of either ChromaTide Alexa Fluor 568–5-dUTP (ThermoFischer Scientific C11399) or ChromaTide Alexa Fluor 488–5-dUTP (ThermoFischer Scientific C11397) in 1x T4 DNA ligase buffer (NEB B0202S) at room temperature overnight. Fluorescently labeled DNA was then purified by ethanol precipitation, resuspended in 1xTE, and dialyzed overnight in 1xTE to remove any residual nucleotides.

DNA was biotinylated by performing fill-in reactions with 5U Klenow exo- fragment (NEB M0212S) and 0.8 mM dTTP, 0.8 mM dGTP, 3.2 μM bio-dCTP, 8 μM bio-dATP (NEB N0446S, Thermo Fisher 19518018, R0081). The reaction was incubated at room temperature overnight and then DNA were purified by ethanol precipitation. Purified DNA were then resuspended in 1xTE to a working concentration of 4 mg/mL.

### Curtain assays

DNA curtain experiments were prepared and executed as described elsewhere (*Gallardo et al., 2015*; *Greene et al., 2010*). Briefly, UV lithography was used to pattern chromium onto a quartz microscope slide, which was then assembled into a flowcell (*Figure 1A*). A lipid bilayer was established within the flowcell by injecting a lipids mix containing 400 μg/mL DOPC, 40 μg/mL PEG-2000 DOPE, and 20 μg/mL biotinylated DOPE (Avanti Polar Lipids 850375, 880130, and 870273) diluted in lipids buffer (100 mM NaCl, 10 mM Tris pH 7.5). Streptavidin, diluted in BSA buffer (20 mM HEPES pH7.5, 70 mM KCl, 20 μg/mL BSA, and 1 mM DTT), was then injected into the flowcell at a concentration of 30 μg/mL. Biotinylated DNA from bacteriophage λ, prepared as described above, was then injected into the flowcell and anchored to the bilayer via a biotin-streptavidin linkage. Buffer flow was then used to align the DNA at the nanofabricated barriers and maintain the curtain in an extended conformation during experiments.

End-labeling of DNA was accomplished using dCas9 molecules. Specifically, dCas9 (IDT 1081066), Alt-R CRISPR-Cas9 tracrRNA (IDT 1072532), and an Alt-R CRISPR-Cas9 crRNA targeting bacteriophage λ at position 47,752 (AUCUGCUGAUGAUCCCUCCG) were purchased from IDT (Integrated DNA Technologies). Guide RNAs were generated by mixing 10 μM crRNA and 10 μM tracrRNA in in Nuclease-Free Duplex Buffer (IDT 11050112), heating to 95C for 5 min and then slowly cooling to room temperature. Guide RNAs were then aliquoted and stored at −20°C.

To prepare Cas9 RNPs for labeling, 200 nM of dCas9 was mixed with 1 μM of guide RNA in dCas9 Hybridization Buffer (30 mM HEPES pH 7.5 and 150 mM KCl) and incubated for 10 min at room temperature. Next, 166 nM of the dCas9-RNA mixture was incubated with 0.08 mg/mL of 6x-His Tag Antibody conjugated with Alexa Fluor 555 (Invitrogen MA1-135-A555) on ice for 10 min. Labeled RNPs were then diluted in BSA buffer and injected into the flowcell at a final concentration of 4 nM. Labeled dCas9 were allowed to incubate with DNA in the flowcell for 10 min before being washed out using imaging buffer (BSA Buffer supplemented with an oxygen scavenging system consisting of 50 nM protocatechuate 3,4-dioxygenase (Fisher Scientific ICN15197505) and 31 μM protocatechuic acid (Abcam ab142937)). Experiments where DNA are labeled, imaging buffer included 20pM YOYO-1 (Thermo Fisher Y3601).

For compaction experiments, HP1 proteins were diluted to the stated concentration in imaging buffer and injected into the flowcell at a rate of 0.7 mL/min. The volume of protein injected was

decided based on protein concentration: for experiments with 50 µM protein, 100 µL was injected, for 5 µM protein, 200 µL was injected, and for 500 nM protein, 400 µL was injected. For experiments utilizing fluorescent HP1, labeled protein was included at the following amounts: 200 nM HP1α−488 was included in the injection 50 µM HP1α, 100 nM HP1α−488 was included in the injection 5 µM HP1α, 400 nM HP1β−488 was included in the injection of 50 µM HP1β, and 400 nM HP1γ−488 was included in the injection of 50 µM HP1γ. After each experiment, HP1 was removed by washing 0.5M KCl, and replicates performed. Data was analyzed as described below.

## Tracking fluorescence during compaction

We measure the fluorescence intensity of both HP1α−488 and YOYO-1 during DNA compaction. For this analysis, individual ROIs of DNA compaction are segmented manually (*Figure 1G*). Data were collected for the average and total fluorescence intensity, mean position, and area of both the compacted and uncompacted segments of the DNA.

## Conservation of YOYO-1 fluorescence

To evaluate our analysis of fluorescence signals due to protein binding, we first tested whether the fluorescence signal from YOYO-1 is conserved across the compacted and uncompacted segments of the DNA during compaction. Assuming YOYO-1 binding is at an equilibrium and uniformly distributed across the DNA, we expect the following to be true: (i) the total intensity of the uncompacted segment, $I_u$, is at a maximum before any compaction begins. (ii) the total intensity of the compacted segment, $I_c$, is at maximum at the end of compaction, and (iii) max $I_u$ = max $I_c$. For the analysis we do the following: (1) at each time frame $I_u$ and $I_c$ are measured. (2) $I_u$ is then fit to a line. (3) The value from the linear fit of $I_u$ is subtracted from $I_c$. And finally (4) $I_c - I_u$ is normalized by dividing through by max $I_c$. The final value, $(I_c - I_u)/\max(I_c)$ follows our expectations spanning $[-1, 1]$ and crossing zero at the midpoint in the compaction process (*Figure 1—figure supplement 2A*).

## Association rates of fluorescent HP1α

To investigate the association of HP1α during compaction, we measure the increase in fluorescent signal along the DNA in both uncompacted and compacted DNA regions. We find on the uncompacted segment of the DNA, the average fluorescence density, $\rho(t)$, increases linearly (*Figure 1I*). For the compacted regions of DNA, the rate of HP1α fluorescence increase is complicated by compaction—the fluorescence can increase both from the association of HP1α from solution and from incorporation of HP1α-bound DNA into the growing compacted segment.

We first consider the increase in fluorescence in the case where HP1α binds to the compacted and uncompacted segments at the same rate. Then, the fluorescence intensity, $I_a$, of the compacted segment at a time, $t$, is given by:

$$I_a(t) = \rho(t)l_c(t)$$

$$I_a(t) = \kappa v t^2$$

where κ is the apparent linear association rate constant measured on uncompacted DNA, $v$ is the linear compaction rate, and $l_c$ is the length of the compacted DNA segment. Alternatively, the fluorescence intensity, $I_b$, when association to previously compacted DNA is blocked, is given by adding up contributions from HP1α association at the time of compaction:

$$I_b(l_c) = \int_0^L \rho(t)dl_c$$

$$I_b(t) = v\kappa \int_0^t tdt$$

$$I_b(t) = \frac{1}{2}\kappa v t^2$$

Thus, if upon condensation, DNA becomes unable to incorporate more HP1α from solution, the

rate of fluorescence increase over a time will lag behind a scenario where HP1α can associate to all of the DNA by a factor of 0.5. A more general model would allow for a variable association rate to the compacted segment that depends on both time and the length of compacted DNA.

$$I(l_c, t) = \int_0^L \rho(t, l_c) dl_c$$

Together, this means that if we normalize our measurement of fluorescence intensity by the expected intensity of equal binding, $\rho(t)l_c(t)$, the resulting trend should center on one in the case of equal binding to both the compacted and uncompacted segments of DNA. Or center on 0.5 if there is no binding to the compacted segment. Generally, a values below one or a negative trendline would indicate that binding to the compacted DNA is impaired relative to uncompacted DNA while values above one or a positive trend would suggest that binding to the compacted segment is enhanced relative to the uncompacted DNA (*Figure 1I–K*).

## Tracking DNA compaction

To track the length of the DNA, we first use an automated program to locate DNA within our images (*Figure 1—figure supplement 2B*). This method is described here: https://github.com/ReddingLab/Learning/blob/master/image-analysis-basics/5__DNA_curtain_finder_1.ipynb.

Once we have the DNA identified, we make kymograms of each individual DNA strand (*Figure 1—figure supplement 2C*). To make the kymograms, we average over the three rows of pixels local to each DNA strand and stack up the average slice across the frames of the video. Then the kymograms are smoothed using a Gaussian filter. After smoothing, we then take the derivative of the image (*Figure 1—figure supplement 2C*). The derivative is generally at its lowest value at the edge of the DNA where the intensity drops off to background levels, and we set the minimum value, by row, of the derivative filtered image as the end position of the DNA (note the directionality of the derivative is top to bottom) (*Figure 1—figure supplement 2C*).

In addition to smoothing the image prior to taking the derivative, we perform two added filtering steps on the data. First, we discount pixels near the edge of the image from the analysis. This is both because often these pixels are added to the kymograms as padding for output and because we know the end of the DNA is not located off image. The second filter is to account for the fact that we expect a relatively smooth trajectory of the DNA end during compaction. To select for this, we take the positions of the DNA end as determined by the minimum of the derivative and apply a Savitzky-Golay filter (SciPy.org). Then, position measurements that are more than a few pixels off this smoothed line are discarded. The general analysis pipeline is automated; however, all fits are manually inspected, and fits deemed to be poor due are removed.

## HP1α-binding site size

The end-to-end distance of an HP1α dimer in the closed conformation is 12.9 nm. The end to end distance of a phosphorylated HP1α dimer phosphorylated in the open conformation is 22.2 nm (*Larson et al., 2017*). Assuming 0.34 nm/bp, we estimate the minimal binding unit of a HP1α dimer in the open conformation is ~65 bp.

## Phasing assays

HP1 condensates were imaged using microscopy grade 384-well plates (Sigma-Aldrich M4437). Prior to use, individual wells were washed with 100 µL of 2% Hellmanex (Sigma-Aldrich Z805939) for 1 hr. Then wells were rinsed three times with water and 0.5M NaOH was added to each well for 30 min before again rinsing three times with water. Next, 100 µL of 20 mg/mL PEG-silane MW-5000 (Laysan Bio MPEG-SIL-5000) dissolved in 95% EtOH was pipetted into each well and left overnight at 4C protected from light. Next, wells were rinsed three times with water and 100 mg/mL BSA (Fisher Scientific BP1600) was pipetted into each well and allowed to incubate for 30 min. Finally, wells were rinsed three times with water and three times with 1x phasing buffer (20 mM HEPES pH 7.5, 70 mM KCl, and 1 mM DTT) was added to each well. Care was taken to maintain 10 µL of volume at the bottom of the well in all steps to prevent drying of the PEG Silane coating of the bottom of the well.

In preparation of experiments, HP1 proteins and DNA substrates were dialyzed overnight into 1x phasing buffer. Then, Protein and DNA were added to a 1.5 mL microcentrifuge tube at 1.5x of the

final concentration stated in results. Excess phasing buffer was removed from cleaned wells and exactly 10 µL of 1x phasing buffer was added to the bottom of the well. Then 20 µL of the protein-DNA solution was then added to the well, resulting in a 30 µL solution of DNA and protein at the concentrations reported in the results section.

In general, all experiments visualizing condensates were performed in triplicate. To generate the phase diagram for HP1α (*Figure 2A*), determine condensate radius (*Figure 2B*, *Figure 2—figure supplement 1*, *Figure 2—figure supplement 2*), and for general condensate assays in *Figure 2D–E*, *Figure 2—figure supplement 2B–C,E*, *Figure 5D–E*, *Figure 6A–B*, *Figure 7F–G,I*, and *Figure 8A*, condensates were visualized by brightfield microscopy at ×20 magnification. Condensates were prepared as described above and allowed to incubate for 1 hr at room temperature before imaging. However, for droplet coalescence assays (*Figure 2D*), droplets were visualized immediately after the reactions were added to the well. The assays in *Figure 3A–F*, *Figure 8B–C*, *Figure 3—figure supplement 1*, and *Figure 3—figure supplement 2* were imaged by spinning disk confocal microscopy at ×100 magnification.

For the mixing assays in *Figure 3D* and *Figure 3—figure supplements 2D*, 100 µM HP1α was mixed with 50 ng/µL 2.7 kbp DNA in 1x phasing buffer for 5 min in two separate reactions with an additional 200 nM HP1α−488 or 200 nM HP1α−565 added to each reaction. Then, a single-color reaction was added to a well, briefly imaged, followed by addition of the remaining reaction. The DNA mixing experiments in *Figure 3E* and *Figure 3—figure supplement 2E* were experiments performed identically to above, except the reactions were prepared using either 50 ng/µL 2.7 kb-488 or 50 ng/µL 2.7 kb-565 and unlabeled protein.

For the MNase assays in *Figure 3F*, condensates were formed by incubating 50 µM HP1α and either 12.5 ng/µL 9 kbp-488 or 12.5 ng/µL 9 kbp-565 for 5 min. Then individual reactions were mixed and incubated at room temperature for 1 hr prior to imaging. MNase digestion was initiated by the addition of 1 mM $CaCl_2$ and 20U MNase (NEB M0247S) and mock reactions were initiated by addition of 1 mM $CaCl_2$ alone.

For the competition experiments in *Figure 8A*, HP1α was first mixed with either HP1β or HP1γ to the stated final concentrations. This solution was then added to 147 bp DNA (250 nM final concentration) and allowed to incubate for 1 hr at room temperature prior to imaging.

For the competition experiments in *Figure 8B–C*, condensates were formed with 50 µM HP1α, 200 nM HP1α−565, and 250 nM 147 bp DNA and incubated for 1 hr at room temperature before briefly imaging. Then, either HP1β−488 or HP1γ−488 was added to the reaction to final concentrations of 50 µM unlabeled protein and 200 nM fluorescent protein.

## Droplet segmentation analysis

Many images of HP1-DNA condensates were collected by brightfield microscopy. Segmenting these droplets presented multiple challenges. For example, the rings of high and low intensity at the edges of the droplets and the fact that the intensity inside droplets is almost the same as background intensity. These factors made analysis with basic threshold segmentation difficult. To overcome these difficulties, we created a custom approach utilizing edge detection and several filters (*Figure 2—figure supplement 2B*). We first high-pass filter the image in Fourier space. Then we detect the edges of condensates with a Canny edge detector (scikit-image.org). Canny edge detection applies a Gaussian filter to smooth the image before taking the gradient. We found that larger condensates were detected more readily when larger values for the variance of the Gaussian filter were used and smaller condensates when smaller values were used. To implement adaptive smoothing, we calculated the edges across a range of sigma values before combining the segments into a single detected image. This method introduced a significant amount of noise. To remove this noise, we utilized two thresholds: one for condensate area (condensates must be larger than three pixels) and the other for condensate eccentricity (condensates must have eccentricity at or less than 0.94).

We segmented at least five separate images for each DNA and protein concentration tested and collected the radius of each detected condensate (*Figure 2B–C*). Then we determined the complementary cumulative distribution (CCD) for condensate radius at each condition (*Figure 2—figure supplement 1*, *Supplementary file 2*). Confidence intervals for each CCD were determined by the Bootstrap method (*Figure 2—figure supplement 2*). Finally, each curve was integrated to determine the expectation value of the radius for each condition (*Figure 2B–C*).

## FRAP assays

For HP1$\alpha$ FRAP experiments, condensates were formed with 100 µM HP1$\alpha$, 250 nM HP1$\alpha$−488, and 50 ng/µL of either linear 147 bp, 2.7 kbp, 9 kbp, or 48.5 kbp DNA (see above, DNA purification). For DNA FRAP experiments, condensates were formed with 100 µM HP1$\alpha$, 100 nM YOYO-1, and 50 ng/µL of either linear 147 bp, 2.7 kbp, 9 kbp, or 48.5 kbp DNA. Samples were then imaged at room temperature (and 5% $CO_2$ for line FRAP experiments). For each photobleaching experiment, automatic focus was activated, pixel binning was set at 2 × 2, and exposure time was set at 300 ms. For line FRAP experiments, a 3 × 512 pixel rectangle was irradiated with 70 mW power at 476 nm (Integrated Laser Engine, Andor) for 300 ms between the 25th and 26th acquired frame. For the whole droplet FRAP experiments, a custom rectangle surrounding a single condensate was irradiated with 70 mW power at 476 nm for 1.5 s between the 10th and 11th acquired frame. Recovery times to half max ($t_{1/2}$) were calculated using a biexponential fit. Data represent three technical replicates of five FRAP experiments, totaling N = 15 for each condition.

## Line FRAP analysis

Line FRAP analysis was performed with a custom R-script. Unbleached condensates, used for normalization, were segmented by threshold. The ROI of bleached regions of condensates (FRAP ROI) was user-defined during imaging. The intensity of the bleached and unbleached condensates as well as background were measured over time. First, the background was subtracted from the FRAP ROI and the unbleached droplets. Then, the FRAP ROI was normalized via the following equation:

$$\frac{I_{FRAP}(t)}{I_{FRAP}(0)} \Big/ \frac{I_{unbleached}(t)}{I_{unbleached}(0)}$$

The normalized intensity was then plotted as a function of time (*Figure 3B*, *Figure 3—figure supplement 2B*) and fit to a bi-exponential fit to determine $t_{1/2}$ values (*Figure 3C*, *Figure 3—figure supplement 2C*).

## Whole drop FRAP

Droplets were formed with 100 µM HP1$\alpha$, 250 nM HP1$\alpha$−488, and 50 ng/µL 2.7 kbp DNA and imaged as described. A square ROI incorporating an entire droplet was photobleached and recovery visualized over 10 min (*Figure 3—figure supplement 1A*).

Unbleached condensates, used for normalization, were segmented by threshold. Due to diffusion and, potentially, the chemical environment of condensates, HP1$\alpha$ fluorescence decays differently inside of droplets relative to background. Therefore, we only use the signal from the fluorescent HP1$\alpha$ within droplets to correct for fluorescence recovery. Additionally, intensity values near the boundary of droplets were omitted from the analysis due to intensity fluctuations resulting from droplet motion. Furthermore, droplets local to the bleached condensate are affected by the bleach strike and are removed from the analysis. Then, we fit the time-dependent decay of condensate fluorescence to a bi-exponential decay equation (*Figure 3—figure supplement 1C-D*).

$$y(t) = ae^{-k_1 t} + be^{-k_2 t}$$

We would then normalize the intensity of the bleached condensate by dividing through by the average decay of unbleached droplets from this equation. However, the intensity of the fluorescent HP1$\alpha$ also decays differently depending on its location within the field of view due to non-homogenous illumination of the sample (*Figure 3—figure supplement 1D*). We therefore scale the decay rates of the unbleached droplets in the following way to correct for spatial variation:

$$\bar{y}(t) = \langle a \rangle e^{-k_1(x,y)t} + \langle b \rangle e^{-k_2(x,y)t}$$

$$k_1(x,y) = k_1^0 + x\alpha_1 + y\beta_1$$

$$k_2(x,y) = k_2^0 + x\alpha_2 + y\beta_2$$

where $\alpha$ and $\beta$ and $k_1^0$ and $k_2^0$ are the slopes and intercepts from a linear regression of decay rate

versus position in the image, $\langle a \rangle$ and $\langle b \rangle$ are the average population factors, and $\bar{y}(t)$ is the adjusted intensity signal.

Next, we use the average corrected rate values from all of the unbleached condensates to normalize the intensity versus time for all the unbleached droplets. We then use the normalized unbleached intensity versus time to visualize the expected spread of the data, which we use as a visual measure of error (*Figure 3—figure supplement 1E*). Finally, we plot the normalized intensity of the bleached condensate against this unbleached distribution to visualize the extent of fluorescence recovery (*Figure 3—figure supplement 1E*).

## Optical trap

Optical trapping experiments were performed on a Lumicks C-Trap G2 system (Lumicks) or a custom-built dual trap. Trapping experiments were performed in specialized flowcells with separate laminar flow channels. For each experiment, two streptavidin-coated polystyrene beads (Spherotec SVP-40–5), diluted in HP1 buffer to 2.2 nM (20 mM HEPES pH 7.5, 70 mM KOAc, 0.2 mg/mL BSA, 1 mM DTT), were captured. Then, the two beads were moved into a channel containing biotinylated λ-DNA diluted to ~0.5 µg/mL in HP1 buffer. Then, using an automated 'tether-finder' routine, a single strand of DNA was tethered between two beads. Each DNA strand was stretched at a rate of 0.1 um per second to a maximal force of 40pN in the buffer-only channel two separate times to measure the force extension curve without HP1 present. Next, trapped DNA molecules were moved to a flow channel containing 10 µM HP1α and 400 nM HP1α−565 and incubated at 5 µm (*Figure 4C–D*, *Figure 4—figure supplement 1A*) or 5.5 µm (*Figure 4E–F*, *Figure 4—figure supplement 1B–E*) extension for 30 s. We then perform stretch-relax cycles (SRC) either with or without waiting periods in the extended or relaxed configurations (*Figure 4—figure supplement 1C*).

For SRCs with no waiting periods (*Figure 4C*, *Figure 4—figure supplement 1A*), we performed fifteen SRCs to a maximal force of 40pN in HP1 buffer with 10 µM HP1α and 400 nM HP1α−565. For SRCs with waiting periods, we performed three consecutive SRCs to a maximal force of 25pN in HP1 buffer with 10 µM HP1α and no additional fluorescent protein. We then moved the DNA tether into a channel containing either HP1 buffer or HP1 buffer supplemented with 500 mM KCl and performed three additional SRCs (*Figure 4—figure supplement 1D–E*).

## Anisotropy

Prior to anisotropy experiments, HP1α, HP1β, and HP1γ were dialyzed overnight into binding buffer (20 mM HEPES pH 7.5, 70 mM KCl, and 1 mM DTT) at 4°C. 60 bp DNA oligos containing a 5'FAM modification (*Supplementary file 1*) were purchased from IDT (Integrated DNA technologies) and diluted to a final concentration of 10 nM in reactions. Binding reactions were then performed in binding buffer supplemented with 0.1 mg/mL BSA and variable amounts of HP1 proteins as indicated. Reactions were incubated for 30 min at room temperature in Corning Low Volume 384-well plates (Corning LCS3821) then measurements were performed on an Analyst HT (Molecular Devices). Data from three independent HP1 titrations were normalized by subtracting the anisotropy value of FAM-60 bp DNA with no added HP1 from each concentration, then fit to a one site binding curve and presented with standard errors.

## Acknowledgements

The authors thank Dr. Emily Wong for her assistance with experiments during the revision process. Special thanks to Lumicks for their generous donation of a C-trap system to the scientific community at the Marine Biological Laboratories during the summer of 2019. The authors would also like to thank Martine Ruer, affiliated to the Hyman lab and the Protein Expression and Purification Facility at the Max-Planck Institute of Molecular Cell Biology and Genetics, Dresden, for the expression and purification of wild-type HP1α for optical tweezers experiments. Special thanks as well to the Biomolecular Nanotechnology Center at the University of California, Berkeley for their assistance in manufacturing flowcells for DNA curtains. We thank Dr. Serena Sanulli and Dr. Adam Larson for helpful comments on the manuscript. MMK was supported by the Discovery Fellows Program at UCSF and NCI grants F31CA243360 and F99CA245719. RR was support from the NOMIS foundation, Rostock, Germany. BH acknowledges support though NIH R21 GM129652, R01 CA231300 and R01 GM131641. BH is also a Chan Zuckerberg Biohub Investigator. SWG was supported by the DFG

(SPP 1782, GSC 97, GR 3271/2, GR 3271/3, GR 3271/4) and the European Research Council (grant 742712). GJN acknowledges support from NIH grant R35 GM127020 and NSF grant 1921794. Support to SR through the UCSF Program for Breakthrough Biomedical Research (PBBR), Sandler Foundation, and Whitman Foundation at the Marine Biological Laboratories.

## Additional information

### Funding

| Funder | Grant reference number | Author |
|---|---|---|
| National Cancer Institute | F31CA243360 | Madeline M Keenen |
| University of California, San Francisco | Discovery Fellows Program | Madeline M Keenen |
| National Cancer Institute | F99CA245719 | Madeline M Keenen |
| NOMIS Stiftung | Graduate Student Fellowship | Roman Renger |
| National Institute of General Medical Sciences | GM129652 | Bo Huang |
| National Institute of General Medical Sciences | GM131641 | Bo Huang |
| National Cancer Institute | CA231300 | Bo Huang |
| Chan Zuckerberg Initiative | Biohub Investigator | Bo Huang |
| Deutsche Forschungsgemeinschaft | SPP 1782 | Stephan W Grill |
| Deutsche Forschungsgemeinschaft | GSC 97 | Stephan W Grill |
| Deutsche Forschungsgemeinschaft | GR 3271/2 | Stephan W Grill |
| Deutsche Forschungsgemeinschaft | GR 3271/3 | Stephan W Grill |
| Deutsche Forschungsgemeinschaft | GR 3271/4 | Stephan W Grill |
| H2020 European Research Council | 742712 | Stephan W Grill |
| National Institute of General Medical Sciences | GM127020 | Geeta J Narlikar |
| National Science Foundation | 1921794 | Geeta J Narlikar |
| University of California, San Francisco | Program for Breakthrough Biomedical Research | Sy Redding |
| Marine Biological Laboratory | Whitman Fellowship | Sy Redding |

The funders had no role in study design, data collection and interpretation, or the decision to submit the work for publication.

### Author contributions

Madeline M Keenen, Conceptualization, Resources, Data curation, Software, Formal analysis, Validation, Investigation, Visualization, Methodology, Writing - original draft, Writing - review and editing; David Brown, Lucy D Brennan, Formal analysis, Investigation, Visualization, Methodology; Roman Renger, Formal analysis, Validation, Investigation, Visualization, Methodology; Harrison Khoo, Resources; Christopher R Carlson, Resources, Investigation; Bo Huang, Resources, Supervision; Stephan W Grill, Resources, Supervision, Methodology; Geeta J Narlikar, Conceptualization, Resources, Supervision, Funding acquisition, Methodology, Writing - original draft, Project administration, Writing - review and editing; Sy Redding, Conceptualization, Resources, Data curation, Software, Formal

analysis, Supervision, Funding acquisition, Investigation, Methodology, Writing - original draft, Project administration, Writing - review and editing

### Author ORCIDs
Bo Huang ⓘ http://orcid.org/0000-0003-1704-4141
Stephan W Grill ⓘ http://orcid.org/0000-0002-2290-5826
Geeta J Narlikar ⓘ https://orcid.org/0000-0002-1920-0147
Sy Redding ⓘ https://orcid.org/0000-0003-3463-7985

### Decision letter and Author response
Decision letter https://doi.org/10.7554/eLife.64563.sa1
Author response https://doi.org/10.7554/eLife.64563.sa2

## Additional files

### Supplementary files
• Supplementary file 1. Tabulated data reported in *Figure 2B–C* and shown in *Figure 2—figure supplement 1*.

• Supplementary file 2. Protein sequences used in this study. Chromodomains (CD) and chromoshadow domains (CSD) are indicated in bold. A 6xHis tag followed by TEV cleavage site tag (MGHHHHHHDYDIPTTENLYFQGS) was appended to each construct for purification.

• Transparent reporting form

### Data availability
The source data and analysis code necessary to reproduce all of the figures included in the manuscript and supporting files are deposited at https://github.com/ReddingLab/Keenen_et_al_bioRxiv_2020.10.30.362772 (copy archived at https://archive.softwareheritage.org/swh:1:rev:77621278d86edccbc226206d892d1f4e358e0fb8/).

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
