## [Decision Letter]

**Acceptance summary:**

This work presents timely new insights into the DNA compaction activity of HP1 and highlights the importance of detailed biophysical studies of phase-separation phenomena to explain and define their biological function. As such, this manuscript will be of interest for anyone studying heterochromatin and/or phase separation in biological processes.

**Decision letter after peer review:**

Thank you for submitting your article "HP1 proteins compact DNA into mechanically and positionally stable phase separated domains" for consideration by *eLife*. Your article has been reviewed by two peer reviewers, one of whom is a member of our Board of Reviewing Editors, and the evaluation has been overseen by Kevin Struhl as the Senior Editor. The reviewers have opted to remain anonymous.

The reviewers have discussed the reviews with one another and the Reviewing Editor has drafted this decision to help you prepare a revised submission.

Summary:

Liquid-liquid phase separation plays an important role in various biological processes. In their manuscript, Keenen et al. perform an elegant in-depth characterization of the in vitro phase separation behavior of the HP1 heterochromatin proteins. The study relies on three main experimental techniques, a single-molecule DNA curtain assay to monitor DNA compaction by the HP1 proteins, an optical tweezer assay to probe the mechanical properties of HP1-DNA complexes, and an ensemble droplet formation assay to observe the mesoscale properties of the HP1 condensates. These techniques enable the authors to make numerous findings regarding the properties of HP1-mediated DNA condensation and phase separation. From their findings, the authors present a model for phase separation by the HP1 proteins and discuss the implications of their model.

This work presents timely new insights into the DNA compaction activity of HP1 and highlights the importance of detailed biophysical studies of phase-separation phenomena to explain and define their biological function. As such, this manuscript will be of interest for anyone studying heterochromatin and/or phase separation in biological processes. We are therefore happy to move forward with the manuscript, provided that the following points are addressed:

Essential revisions:

1) Can the authors comment on how the concentrations of HP1 proteins used in their assays compares to the estimated concentration of the HP1 proteins in the nucleus? If the HP1 protein concentrations required to see phase separation in vitro are substantially larger, could this point to a role for the chromodomain for increasing the local concentration of domains HP1 to enable DNA condensation and phase separation at heterochromatin?

2) Related to the point above, in both this work and other work on phase separation (especially those involving protein-nucleic acid interactions), salt concentrations are very important. The authors perform both the single-molecule and ensemble assays in 70 mM KCl, which is lower than reported physiological KCl concentrations (100-150mM monovalent cations), which makes it difficult to extrapolate their results to the behavior of the systems in vivo. The authors should comment on this.

3) In the phase separation experiments, the authors use the Widom 601 sequence as the 147 bp DNA sequence. This sequence displays unusual properties compared to native DNA sequences in the genome (especially compared to the linker DNAs that would typically be expected to be bound by the HP1 proteins). It would be helpful if the authors could comment on whether they can obtain similar results with a different DNA sequence.

4) In their model, the authors assume that HP1a oligomerization occurs only in the presence of DNA. However, the authors also show that, at sufficiently high concentrations, HP1a can phase separate in the absence of DNA, indicative of some level of self-association. At concentrations where HP1a phase separation is DNA-dependent, do the authors have evidence of whether or not HP1a is capable of oligomerization (e.g., by size-exclusion chromatography and/or dynamic light scattering data)?

---

## [Author Response]

Essential revisions:1) Can the authors comment on how the concentrations of HP1 proteins used in their assays compares to the estimated concentration of the HP1 proteins in the nucleus? If the HP1 protein concentrations required to see phase separation in vitro are substantially larger, could this point to a role for the chromodomain for increasing the local concentration of domains HP1 to enable DNA condensation and phase separation at heterochromatin?

These are excellent points that we agree are important for proper interpretation of our results. We have updated the main text to explicitly state the in vivo concentrations of HP1α. The concentration of HP1α in vivo is estimated between 1.0-10.0μM (Lu et al., 2000; Müller et al., 2009) through studies in *Drosophila* larvae, mouse NIH 3T3 cells and mouse embryonic fibroblasts. To our knowledge, the nuclear concentrations of HP1β and HP1γ have not been investigated at as detailed a level as for HP1α. However from western blots and immunostaining of nuclear fractions, it appears that HP1γ levels are similar to HP1α, while HP1β may be considerably less (Bártová et al., 2005). However, because of the lack of concrete values for HP1γ and HP1β, we only refer to their concentrations relatively in the Discussion. Importantly, it remains possible that the absolute and relative concentrations of the three paralogs are also cell-type dependent.

Importantly, our measurement of the critical concentration for condensate formation by HP1α and longer DNA molecules spans the range of HP1α concentrations measured in vivo. This means, as the reviewers have pointed out, that several factors that can affect HP1α-chromatin interactions are poised to regulate condensation. For example, increasing salt (see below) increases the required concentration of HP1α necessary for condensation. In this context, as the reviewers mentioned, H3K9 methylation recognition could increase HP1α’s affinity for chromatin and raise its local concentration. This overlooked but extremely important idea has now been integrated into the text.

2) Related to the point above, in both this work and other work on phase separation (especially those involving protein-nucleic acid interactions), salt concentrations are very important. The authors perform both the single-molecule and ensemble assays in 70 mM KCl, which is lower than reported physiological KCl concentrations (100-150mM monovalent cations), which makes it difficult to extrapolate their results to the behavior of the systems in vivo. The authors should comment on this.

This is a very pertinent question, and we can speculate as to what we might expect at higher concentrations of KCl. In general, we expect a monotonic decrease in compaction rate, and a corresponding increase in critical concentration as a result of increasing the concentration of KCl. These trends may not be linear but will continue to the point that both compaction and condensation are eventually eliminated. To this point, we have previously shown (Larson et al., 2017), and used in this study (Materials and methods), the fact that HP1α can be removed from DNA curtains by a salt wash of 500mM NaCl. Moreover, we show in Figure 4—figure supplement 1, that 500mM NaCl also abolishes HP1α condensation and mechanical resistance in optical trapping experiments (Results).

To experimentally address the issue raised by the reviewers we have added new data showing the effect of 150mM KCl on HP1α-condensation with 2.7kbp DNA (Figure 2—figure supplement 2E). In these experiments, we show that the critical concentration increases to ~50μM HP1α, which may indicate that HP1α-condensation requires additional positive interactions with nucleosomes or chromatin associated factors to drive condensation of chromatin in vivo. This experiment and its implications have been added to the main text and Discussion.

In general, we recognize that buffer conditions can have profound effects on protein-DNA interactions and the material properties of condensates, especially condensates containing long ionic polymers. In our experiments, we chose to use 70mM KCl in large part because of the experiments themselves. These conditions allowed us to maximize the dynamic range of observables and compare HP1 paralogs using complementary experimental approaches under the same buffer conditions. From the vantage of the results of these experiments, it is clear that for future studies there is real value in a more expansive survey of reaction conditions, including concentration, valency and type of ions, pH, and crowding agents to reveal in greater detail the potential of HP1 proteins to organize DNA.

3) In the phase separation experiments, the authors use the Widom 601 sequence as the 147 bp DNA sequence. This sequence displays unusual properties compared to native DNA sequences in the genome (especially compared to the linker DNAs that would typically be expected to be bound by the HP1 proteins). It would be helpful if the authors could comment on whether they can obtain similar results with a different DNA sequence.

The reviewers bring up a great point. The 601 sequence was originally isolated due to its propensity to bend around the histone octamer. Moreover, disordered polyelectrolytes tend to bend DNA as a means toward compaction. Thus, it may be that HP1α interactions that lead to compaction are more preferred with the Widom 601 sequence than a randomized sequence. However, we do not observe effects that would be expected from robust binding site bending on DNA curtains (Figure 1I, Figure 1—figure supplement 2). And generally, we expect any specific interaction energy, based on DNA sequence, to be overshadowed under the conditions studied here for the following reasons.

First, observations of fluorescent HP1 interactions on DNA from λ-phage do not reveal significant sequence specificity. λ-DNA has several interesting sequence features that might elicit novel binding modes: A GC-rich half, an AT-rich half, and a couple polyA-polyT tracts near its midpoint. Importantly these polyA-polyT tracts inhibit nucleosome assembly and share sequence elements with nucleosome free regions and some linker DNA sequences. Overall λ-DNA has been shown contain sequence features that both promote and resist the formation nucleosomes in vitro (Field et al., 2008; Visnapuu and Greene, 2009). Yet, we do not observe significant or sequence-dependent heterogeneity of HP1 binding by fluorescence signal on the uncompacted regions of λ DNA (Figure 1F). These observations indicate that if any sequences preferences exist, their energetic contributions are likely small in comparison to the strength of non-specific binding interactions under the conditions tested here.

Second, we surveyed several DNA substrates throughout the experiments herein, including the Widom 601 sequence, puc19, a derivative of pET15, DNA from λ-phage, as well as the oligos listed in Table 1. Across these substrates, and all of these sequences they represent, the behavior of each HP1 paralog behaved as expected for non-specific interactions and any specific binding interactions were not obvious in our assays. However, in light of the conversation above concerning salt concentration, it is certainly a formal possibility that certain combinations of ions or pH might amplify or reveal sequence specificity.

4) In their model, the authors assume that HP1a oligomerization occurs only in the presence of DNA. However, the authors also show that, at sufficiently high concentrations, HP1a can phase separate in the absence of DNA, indicative of some level of self-association. At concentrations where HP1a phase separation is DNA-dependent, do the authors have evidence of whether or not HP1a is capable of oligomerization (e.g., by size-exclusion chromatography and/or dynamic light scattering data)?

We thank the reviewers for bringing this point to our attention, because we did not intend to convey that oligomerization occurs *only* in the presence of DNA. Rather, we mean to include a model where higher order oligomerization (oligomerization beyond dimers), can be promoted by DNA binding, which would be consistent with our results. We propose such a model because of the following two observations: (i) In this work we show that at 40mM KCl, HP1α can phase-separate in the absence of DNA at a concentration of ~400uM (Figure 2—figure supplement 2C) (ii) We’ve previously shown that at 75mM KCl, HP1α does not phase-separate on its own even at a concentration of 800uM (Larson et al., Figure 1E). In this previous work we also found that at 75mM KCl, at concentrations where HP1α can phase-separate in the presence of DNA, HP1α on its own, is predominantly a dimer, as measured by Analytical Ultracentrifugation and Multi-angle light scattering (Larson et al., 2017: Figure 2A and Extended Data Figure 6). Based on these findings we propose that HP1α has the ability to form higher-order oligomers by itself, but this is a salt-dependent process that can be also enhanced by the presence of DNA. The manuscript has been updated to reflect the above rationale and hypothesis.

A measurement of higher order oligomerization of HP1α under conditions where it phase-separates (40mM KCl or 75mM KCl +DNA) has proved technically difficult because this requires uncoupling higher order oligomerization from phase-separation so that methods like dynamic light scattering can unambiguously assess soluble oligomerized states as opposed to phase-separated states. To date our attempts at finding conditions and concentration regimes that allow such uncoupling have been unsuccessful. Therefore to this point, we have tried to be very clear that HP1 oligomerization in the presence of DNA is a potential mechanism that could account for many of our measurements of HP1-DNA interactions, but without direct evidence in our experiments. Throughout the text now, we have qualified instances of oligomerization to indicate this point.